# DeepCrossAttention: Supercharging Transformer Residual Connections

**Mike Heddes** [1]  **Adel Javanmard** [2][3]  **Kyriakos Axiotis** [3]  **Gang Fu** [3]  **MohammadHossein Bateni** [3]
**Vahab Mirrokni** [3]

## Abstract

Transformer networks have achieved remarkable success across diverse domains, leveraging a variety of architectural innovations, including residual connections. However, traditional residual connections, which simply sum the outputs of previous layers, can dilute crucial information. This work introduces DeepCrossAttention (DCA), an approach that enhances residual learning in transformers. DCA employs learnable, input-dependent weights to dynamically combine layer outputs, enabling the model to selectively focus on the most relevant information in any of the previous layers. Furthermore, DCA incorporates depth-wise cross-attention, allowing for richer interactions between layers at different depths. Our language modeling experiments show that DCA achieves improved perplexity for a given training time. Moreover, DCA obtains the same model quality up to 3x faster while adding a negligible number of parameters. Theoretical analysis confirms that DCA provides an improved trade-off between accuracy and model size when the ratio of collective layer ranks to the ambient dimension falls below a critical threshold.

## 1. Introduction

Residual connections play an important role in modern neural network architectures because they stabilize the training of deep neural networks and improve model convergence and quality. Since their usage in the ResNet architecture (He et al., 2016), residual connections have been widely adopted in both convolutional neural networks and transformer architectures across various domains, including natural language processing (Vaswani, 2017), audio recognition (Gong et al., 2021), and computer vision (Dosovitskiy et al., 2021).

A residual neural network (ResNet) is constructed by stacking layers known as residual blocks. Each residual block is characterized by the recursive equation $\boldsymbol{x}_{t+1} = f(\boldsymbol{x}_t) + \boldsymbol{x}_t$, which contains a residual function $f$ along with an identity shortcut (also called an identity loop or skip connection). The residual functions typically used in these blocks include multi-layer perceptrons (MLPs), convolutional neural networks (CNNs), and attention. By unrolling the recursion, we equivalently see that each layer's input is the sum of all its previous layers' outputs (including the model's input). Figure 2 provides a schematic illustration of this concept.

**Information dilution in residual networks.** Residual connections increase the flow of information across the neural network. However, they also come with a potential limitation: Taking a straight sum of previous layer outputs implicitly treats all previous layers as equally important. This can dilute useful information present in a select few layers (including the model's input) with potentially less useful information. We hypothesize that, because of this dilution, even though residual networks mitigate the problem of neural network bottlenecks, they do not sufficiently resolve it. One way to resolve the issue of dilution would be to allow each layer to *choose its inputs*.

In order to confirm the existence and significance of the dilution phenomenon we ask a simple question: *Can residual networks easily learn to recover the input?* This should be a basic task expected of any generative model — otherwise there would be information loss. However, if our dilution hypothesis is true, the answer would be negative. To test this, we create a neural network consisting of a number of low-rank layers, and add residual connections in order to mitigate the bottlenecks introduced by the low ranks. The resulting model is full-rank. We compare this model with another model that employs *learnable* residual connections, as in DenseFormer (Pagliardini et al., 2024), which we later also call *GRN-v1*, since it is the starting point of our generalizations. In Figure 1 we see the results of the two models on two tasks: learning the identity transformation and learning a random linear transformation. Perhaps surprisingly, we observe that the residual network is unable

[1]Department of Computer Science, University of California, Irvine, USA [2]Marshall School of Business, University of Southern California, Los Angeles, USA [3]Google Research, New York, USA. Correspondence to: Mike Heddes <mheddes@uci.edu>, Gang Fu <thomasfu@google.com>.

*Proceedings of the 42$^{nd}$ International Conference on Machine Learning*, Vancouver, Canada. PMLR 267, 2025. Copyright 2025 by the author(s).

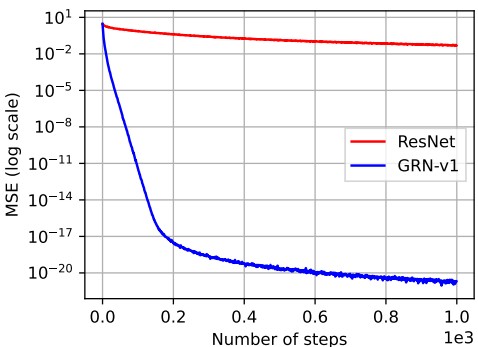
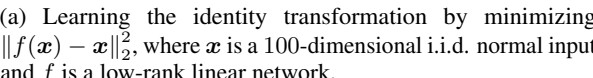

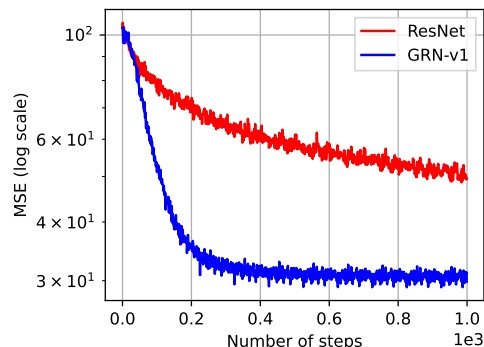

(a) Learning the identity transformation by minimizing $\|f(\boldsymbol{x}) - \boldsymbol{x}\|_2^2$, where $\boldsymbol{x}$ is a 100-dimensional i.i.d. normal input and $f$ is a low-rank linear network.

(b) Minimizing the loss $\|f(\boldsymbol{x}) - \boldsymbol{y}\|_2^2$, where $\boldsymbol{x}$ is a 100-d i.i.d. normal input, $f$ is a low-rank linear network, and $\boldsymbol{y} = \boldsymbol{Ax} + \boldsymbol{b}$, where $\boldsymbol{A}, \boldsymbol{b}$ have i.i.d. standard normal entries.

*Figure 1.* Training low-rank linear models to learn the identity and a random transformation. Each model consists of 10 linear layers, each of rank 3, and is trained using mini-batch SGD.

to fully reconstruct the input even after seeing $10^3$ batches ($10^5$ examples), while the model with learnable residual weights is able to reach extremely small loss values, even with 100x fewer examples. This confirms that ResNet does not address neural network bottlenecks in a satisfactory way, even though it learns a full-rank transformation, and underscores the importance of using learnable residual weights to increase model capacity.

**Our contribution.** In this work, we propose *DeepCrossAttention (DCA)*, a new transformer architecture that generalizes residual networks by employing learnable, input-dependent weights to dynamically combine layer outputs, enabling the model to selectively focus on the most relevant information in any of the previous layers and thereby prevent dilution of information in the hidden representations. Furthermore, DCA incorporates depth-wise cross-attention by enabling the queries, keys, and values in each transformer block to independently combine layer outputs, allowing for richer interactions between layers at different depths. This is all achieved with a negligible number of additional parameters, making DCA more effective than increasing the model size (for instance by increasing its width or depth).

DCA can be viewed as a mechanism to adapt the model architecture dynamically for each input token. By optimizing the added parameters, DCA learns to effectively combine the outputs of earlier residual blocks. This allows the model to rearrange the residual blocks from purely sequential to fully parallel and any intermediate combination, without the need for explicit architectural design choices.

We analyse our generalization of the residual network theoretically by focusing on a linear low-rank model. We show that DCA achieves a better trade-off between accuracy and model size when the ratio of the collective ranks of the lay-

ers to the ambient dimension is below a threshold, which depends on the complexity of the target task. In addition, the improvement in this trade-off can itself be characterized as a function of the collective ranks of the layers, ambient dimension and the complexity of the target task. We extend this insight to nonlinear models by working with the notion of bottleneck rank, proposed by Jacot (2023).

We additionally provide empirical results to support the theoretical findings and demonstrate the effectiveness of DCA. Experiments on language modeling and image classification tasks demonstrate that DCA consistently outperforms the standard transformer architectures in terms of perplexity, accuracy and training efficiency. DCA achieves lower perplexity for a given parameter budget and training time. Furthermore, DCA exhibits improved training stability, mitigating the occurrence of loss spikes frequently observed while training large models.

## 2. Related Work

Highway networks enable each layer to interpolate dynamically between its output $f(\boldsymbol{x})$ and its input $\boldsymbol{x}$ using a gating mechanism (Srivastava et al., 2015). Residual connections (He et al., 2016) popularized the direct flow of information from earlier to later layers using identity shortcuts. These innovations proved crucial in stabilizing training and allowing for the construction of significantly deeper networks. Building upon this concept, DenseNet (Huang et al., 2017) further enhanced information flow by concatenating the outputs of all preceding layers to each layer's input.

The following methods are the ones most similar to ours. They all build on the idea of DenseNet but apply an efficient aggregation of the previous layer outputs instead of

concatenating them. DenseFormer (Pagliardini et al., 2024) performs the aggregation as a learned linear combination of the previous layer outputs. To reduce the computational load, they propose to apply their method only on a subset of the possible layer connections. Building on DenseFormer, LAuReL (Menghani et al., 2024) presents three aggregation functions, the best performing one applies a learned low-rank transformation to the previous layer outputs before the learned linear combination. Zhu et al. (2024) take a different approach with Hyper-Connections, they consider a fixed-size stack where layer outputs are added into with a learned weight for every slot of the stack. Before each layer, the stack is mixed by a matrix multiplication with a learned weight matrix. The input to a layer is then obtained by a learned linear combination of the stack, instead of accessing the previous layer outputs directly. They also present a dynamic version of their method where the weights are derived from the inputs.

## 3. Method

We start with a detailed exposition of our proposed generalizations to the residual network architecture. We present three distinct proposals, each incrementally augmenting the complexity of the network structure. Building upon these proposals, we subsequently introduce DeepCrossAttention (DCA), a novel approach to enhance residual learning capabilities of the transformer architecture.

**Notation.** We denote a residual function by $f_t : \mathbb{R}^d \to \mathbb{R}^d$, where $t$ is the layer index and $d$ the feature dimension. As an example, in a multi-layer perceptron residual network (MLP-ResNet), we have $f_t(\boldsymbol{x}) = \boldsymbol{V}_t \sigma(\boldsymbol{W}_t \boldsymbol{x})$ with $\boldsymbol{W}_t \in \mathbb{R}^{k \times d}$, $\boldsymbol{V}_t \in \mathbb{R}^{d \times k}$ and $\sigma$ is a nonlinear function, such as sigmoid or ReLU, that is applied component-wise. Then, the $t$-th residual block outputs $g_{t+1}(\boldsymbol{x})$, are defined recursively as

$$g_{t+1}(\boldsymbol{x}) = f_t(g_t(\boldsymbol{x})) + g_t(\boldsymbol{x}) \,.$$

Using this recursion, the output of the $T$-th residual block is given by

$$g_{T+1}(\boldsymbol{x}) = \sum_{t=0}^{T} f_t(g_t(\boldsymbol{x})) \,,$$

with the conventions that $g_0(\boldsymbol{x}) = \boldsymbol{0}$ and $f_0(g_0(\boldsymbol{x})) = \boldsymbol{x}$. We refer to Figure 2 for a schematic illustration.

An alternative description, which we will use to introduce our generalizations, is the following. For every $t$, define the stack of layer outputs $\boldsymbol{G}_t \in \mathbb{R}^{d \times t}$ as

$$\boldsymbol{G}_t := \left[ f_{t-1}(g_{t-1}(\boldsymbol{x})), \dots, f_0(g_0(\boldsymbol{x})) \right] \in \mathbb{R}^{d \times t} \,.$$

We then have $g_t(\boldsymbol{x}) = \boldsymbol{G}_t \mathbf{1}$ and $\boldsymbol{y} = \boldsymbol{G}_T \mathbf{1}$ in the standard residual network, where $\mathbf{1}$ denotes the all ones vector.

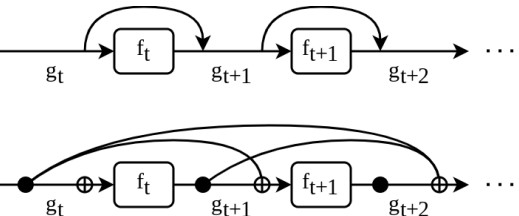

*Figure 2.* Two alternative schematic representations of standard ResNet. The top represents the recursive form, the bottom represents the explicit sum.

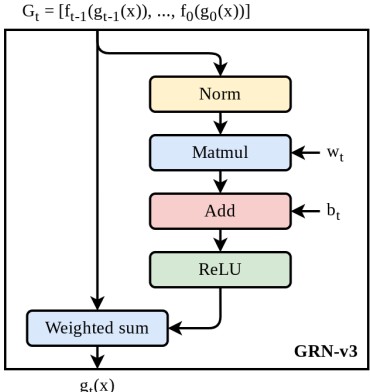

*Figure 3.* Computation diagram of GRN-v3.

### 3.1. Generalized Residual Networks (GRN)

We propose three generalizations of ResNets by considering weighted linear combinations of previous layer outputs. The parameters of the modules and the generalizations are all optimized during training using the AdamW optimizer (Loshchilov & Hutter, 2017).

**Dimension-independent weights (GRN-v1).** We consider simple linear combinations as

$$g_t(\boldsymbol{x}) = \boldsymbol{G}_t \boldsymbol{b}_t, \quad \boldsymbol{y} = \boldsymbol{G}_{T+1} \boldsymbol{b}_{T+1}$$

with $\boldsymbol{b}_t \in \mathbb{R}^{t \times 1}$ which is initialized as all ones and optimized with the rest of the model parameters during training. This setting has been previously explored in the Dense-Former paper (Pagliardini et al., 2024).

**Dimension-dependent weights (GRN-v2).** In this proposal, we allow $\boldsymbol{b}_t \in \mathbb{R}^{d \times t}$ and consider

$$g_t(\boldsymbol{x}) = (\boldsymbol{G}_t \odot \boldsymbol{b}_t)\mathbf{1}, \quad \boldsymbol{y} = (\boldsymbol{G}_{T+1} \odot \boldsymbol{b}_{T+1})\mathbf{1} \,,$$

where $\odot$ indicates the entry-wise (Hadamard) product. Note that in GRN-v1 the same weight vector $\boldsymbol{b}_t$ is used for each of the $d$ features. GRN-v2 generalizes this by using different weight vectors for different features, which are all stacked together in a matrix $\boldsymbol{b}_t \in \mathbb{R}^{d \times t}$.

**Input-dependent weights (GRN-v3).** In the next generalization, we allow the weights to be input dependent. Specifi-

cally, the weights are given by $\boldsymbol{b}_t + \bar{\boldsymbol{w}}_t$ with $\boldsymbol{b}_t, \bar{\boldsymbol{w}}_t \in \mathbb{R}^{d \times t}$. The first component acts similar to the weights in GRN-v2, it puts different weights on different dimensions of the input. The second component $\bar{\boldsymbol{w}}_t$ is a nonlinear mapping of the input features vector $\boldsymbol{x}$, but is the same for all the $d$ dimensions. This combination gives us flexibility to have both dimension-dependent and input-dependent weights for a slight increase in the number of parameters. GRN-v3 is expressed as

$$g_t(\boldsymbol{x}) = (\boldsymbol{G}_t \odot (\boldsymbol{b}_t + \bar{\boldsymbol{w}}_t))\mathbf{1}, \quad \bar{\boldsymbol{w}}_t = \mathbf{1}\sigma(\boldsymbol{w}_t^\mathsf{T}\boldsymbol{G}_t),$$
$$\boldsymbol{y} = (\boldsymbol{G}_{T+1} \odot (\boldsymbol{b}_{T+1} + \bar{\boldsymbol{w}}_{T+1}))\mathbf{1}$$

where $\boldsymbol{w}_t : \mathbb{R}^{d \times 1}$ is initialized as all zeros and optimized with the rest of the model parameters during training and $\sigma : \mathbb{R} \to \mathbb{R}$ is a non-linearity which is applied entry-wise. In this proposal we consider $\sigma$ to be the ReLU activation. The computation diagram of GRN-v3 is illustrated in Figure 3.

**Reducing memory and computation.** Since the stack of layer outputs $\boldsymbol{G}_t$ grows linearly with the depth of the model, this could lead to significant memory and computational overhead for deep models. Our experiments reveal that GRNs tend to weight inputs and the last few layer outputs the most. An example weight distribution is provided in Appendix H. Therefore, to increase efficiency, we propose to include only the first and last-$k$ layers explicitly in $\boldsymbol{G}_t$. On the intermediate layers we apply standard ResNet, only involving simple addition. For example, if we set $k = 2$, then $\boldsymbol{G}_t$ contains at most 4 vectors: the model inputs, the sum of the intermediate layers' outputs, and the last two layers' outputs $f_{t-1}(g_{t-1}(\boldsymbol{x}))$ and $f_{t-2}(g_{t-2}(\boldsymbol{x}))$. The GRNs then take this modified $\boldsymbol{G}_t$ as their input.

### 3.2. DeepCrossAttention

The generalizations introduced thus far are generally applicable to any ResNet. We now describe our main method which is specific to the transformer architecture. DeepCrossAttention (DCA) generalizes self-attention by adding three independent instances of a GRN in each decoder block. In this proposal we consider the GRN to be GRN-v3. These three GRN instances are given the same stack of previous layer outputs as their input but return the queries, keys, and values for the attention module, respectively. This enables richer interactions between layers at different depths. Figure 4 shows the computation diagram of a DCA decoder block inside a transformer, where the remaining skip connections ensure that the inputs are not added to the outputs of the decoder block, but are included in the inputs of both the attention and the feed forward module. Notably, DCA does not modify the underlying attention mechanism, but instead uses GRNs to dynamically compose attention inputs.

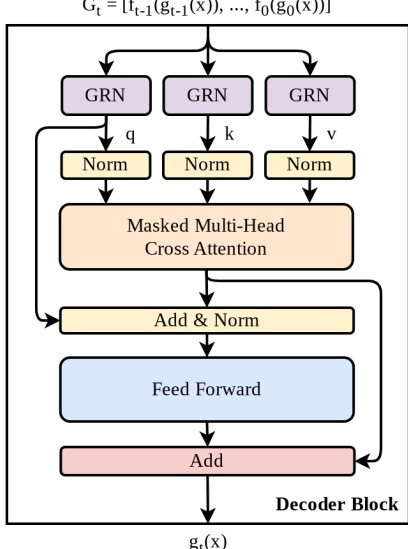

*Figure 4.* Computation diagram of a DCA decoder block.

## 4. Theoretical analysis

Motivated by language modeling tasks, we focus on the regime where the size of the training set ($n$) significantly exceeds the input dimension ($n \gg d$). As we increase the number of model parameters, the representation capacity of the network improves, which helps with reducing the test error. We will be focusing on the the trade-off between the test error and the number of parameters, and argue that our proposed generalizations achieve a better trade-off than the standard ResNet.

We will first study a "stylized" low-rank linear model for which we characterize the test error-model complexity trade-off and demonstrate the benefits of our proposed generalizations. Our analysis elucidates the role of various factors on this trade-off, such as collective widths of layers, complexity of the target task, and input dimension. We then discuss how some of these results can be extended to non-linear models and empirically demonstrate that the insights gained from our analysis are applicable to more complex models.

Due to space constraint, proof of theorems are deferred to the supplementary material.

### 4.1. Low-rank linear model

Consider the setting where for each sample the response $\boldsymbol{y} \in \mathbb{R}^d$ is given by

$$\boldsymbol{y} = \boldsymbol{A}\boldsymbol{x} + \epsilon$$

with $\epsilon \in \mathbb{R}^d$ representing the noise. Here $\boldsymbol{A} \in \mathbb{R}^{d \times d}$ is a full rank matrix.

We consider a network with $T$ layers where $f_t(\boldsymbol{z}) = \boldsymbol{V}_t$

(there is no activation). We let $r_t := \text{rank}(V_t)$ and define the collective rank $r_* := \sum_{t=1}^T r_t$. We assume $r_* < d$, i.e., the collective rank of all layers still is lower than the ambient dimension $d$.

We next focus on four architectures: Baseline (where there is no residual connection), ResNet, GRN-v1 and GRN-v2 and characterize the class of models which can be expressed by each of these architectures. We assume each architecture to have $T$ layers.

**Baseline.** In this architecture, there is no residual connection and so the model is given by $\widehat{y} = \prod_{t=1}^T V_t x$. We denote by $\mathcal{C}_{\text{base}}$ the class of functions that can be represented by such architecture.

**ResNets.** In this case, we have $\widehat{y} = \prod_{t=1}^T (I + V_t)x$. Denote by $\mathcal{C}_{\text{res}}$ as the class of functions that can be represented by such architecture.

**GRN-v1.** In this case, we have $\widehat{y} = G_{T+1}b_{T+1}$, with $b_{T+1}$ a $(T+1)$-dimensional vector as described in Section 3. Denote by $\mathcal{C}_{\text{GRN-v1}}$ the class of functions that can be represented by such architecture.

**GRN-v2.** In this case, we have $\widehat{y} = (G_{T+1} \odot b_{T+1})\mathbf{1}$, where $b_{T+1}$ is $d \times (T+1)$ matrix as described in Section 3. We denote by $\mathcal{C}_{\text{GRN-v2}}$ the class of functions that can be represented by such architecture.

**GRN-v3.** In this case, we have $\widehat{y} = (G_{T+1} \odot (b_{T+1} + \bar{w}_{T+1}))\mathbf{1}$, where $b_{T+1}$ is $d \times (T+1)$ matrix and $\bar{w}_{T+1}$ is $d$ dimensional vector as described in Section 3. We denote by $\mathcal{C}_{\text{GRN-v3}}$ the class of functions that can be represented by such architecture.

**Theorem 4.1.** *For the low rank linear model we have:*

- $\mathcal{C}_{\text{base}} = \{x \mapsto Mx : \text{rank}(M) \leq \min(r_t)_{t=1}^T\}$.

- $\mathcal{C}_{\text{res}} = \{x \mapsto (I + M)x : \text{rank}(M) \leq r_*\}$.

- $\mathcal{C}_{\text{GRN-v1}} = \{x \mapsto (\alpha I + M)x : \text{rank}(M) \leq r_*\}$.

- $\mathcal{C}_{\text{GRN-v2}} = \{x \mapsto (D + M)x : \text{rank}(M) \leq r_*, D \text{ is diagonal}\}$.

- $\mathcal{C}_{\text{GRN-v3}} \supset \{x \mapsto (D + M)x + \sigma(w^\mathsf{T}x)x : \text{rank}(M) \leq r_*, D \text{ is diagonal}, w \in \mathbb{R}^{d \times 1}\}$.

### 4.2. Trade-off between test error and model complexity

In the previous section, we characterized the class of models that can be expressed by each architecture. Next, we study the trade-off between the optimal test error achievable by each model and the model complexity, defined as the number of its parameters.

Note that all the classes of models characterized in Theorem 4.1 are linear functions. For a linear model $x \mapsto \widehat{A}x$,

its test error (model risk) is given by

$$\text{Risk}(\widehat{A}) = \mathbb{E}[(y - \hat{y})^2]$$
$$= \mathbb{E}\left[\left\|(A - \widehat{A})x\right\|_{\ell_2}^2\right] + \sigma^2$$
$$= \mathbb{E}[\text{trace}\{(A - \widehat{A})xx^\mathsf{T}(A - \widehat{A})^\mathsf{T}\}] + \sigma^2$$
$$= \left\|A - \widehat{A}\right\|_F^2 + \sigma^2,$$

where we assumes that $\mathbb{E}[xx^\mathsf{T}] = I$ (isotropic features). Since the term $\sigma^2$ is constant (independent of model $\widehat{A}$) we will drop it in sequel without effecting our discussion and focus on the excess risk. For a class of models $\mathcal{C}$ we use the notation $\text{ER}^*(\mathcal{C})$ to indicate the minimum excess risk achievable over the class $\mathcal{C}$:

$$\text{ER}^*(\mathcal{C}) := \min_{\widehat{A} \in \mathcal{C}} \left\|A - \widehat{A}\right\|_F^2.$$

Note that $\text{ER}^*(\mathcal{C}_{\text{base}}(T))$ is obtained by the best $r$-rank approximation to $A$ and $\text{ER}^*(\mathcal{C}_{\text{res}})$ is obtained by the best $rT$-rank approximation to $A - I$, both of which have simple characterization in terms of the singular values of $A$ and $A - I$, by using the celebrated Eckart–Young–Mirsky theorem. Deriving $\text{ER}^*(\mathcal{C}_{\text{GRN-v1}}(T))$ and $\text{ER}^*(\mathcal{C}_{\text{GenB(T)}})$ are more complicated. In the next theorem, we establish upper bounds on them.

**Theorem 4.2.** *Consider the singular value decomposition $A - I = U\Sigma V^\mathsf{T}$. For a given $m \in [d]$, let $U_m$, $\Sigma_m$, $V_m$ be the top $m$ singular vectors and singular values and define $\Delta := A - I - U_{r_*}\Sigma_{r_*}V_{r_*}^\mathsf{T}$, where $r_* := \sum_{\ell=1}^T r_\ell$. We then have*

$$\text{Err}^*(\mathcal{C}_{\text{res}}) = \|\Delta\|_F^2,$$

$$\text{Err}^*(\mathcal{C}_{\text{GRN-v1}}) \leq \|\Delta\|_F^2 - \frac{1}{d - r_*}\text{trace}(\Delta)^2,$$

$$\text{Err}^*(\mathcal{C}_{\text{GRN-v2}}) \leq \|\Delta\|_F^2$$
$$- \max\left\{\sum_{i=1}^d \Delta_{ii}^2, \frac{1}{d - r_*}\text{trace}(\Delta)^2\right\},$$

*where $\{\Delta_{ii}\}_{i=1}^d$ are the diagonal entries of $\Delta$. In addition,*

$$\text{Err}^*(\mathcal{C}_{\text{GRN-v3}}) \leq \|\Delta\|_F^2 - \max\{\mathcal{T}_1, \mathcal{T}_2\},$$
$$\mathcal{T}_1 := \frac{1}{d - r_*}\text{trace}(\Delta)^2 + \frac{\nu_{\max}^2}{\pi(d+1)},$$
$$\mathcal{T}_2 := \sum_{i=1}^d \Delta_{ii}^2 + \frac{\tilde{\nu}_{\max}^2}{\pi(d+1)}.$$

*Here $\nu_{\max}$ denotes the maximum eigenvalue of $\frac{\Delta + \Delta^\mathsf{T}}{2} - \frac{1}{d - r_*}\text{trace}(\Delta)U_{r_*,\perp}U_{r_*,\perp}^\mathsf{T}$ and $\tilde{\nu}_{\max}$ denotes the maximum eigenvalue of $\frac{\Delta + \Delta^\mathsf{T}}{2} - diag(\Delta)$.*

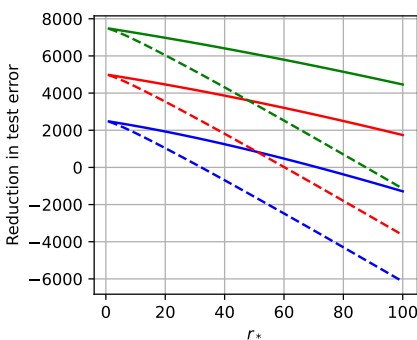 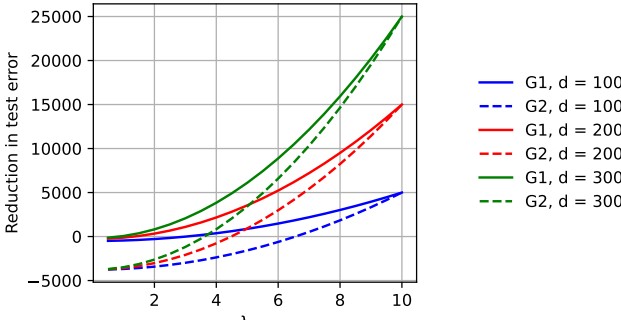

Figure 5. Gain in the model performance achieved by GRN-v1 and GRN-v2 over ResNet. The plots represents the lower bounds for $G_1$ and $G_2$ given in Theorem 4.3(ii). Observe that the gain at larger dimension $d$ is higher. Left panel shows that the gain decreases as the collective rank $r_*$ of ResNet increases ($\lambda_{\min} = 5$, $\lambda_{\max} = 10$). Right panel shows that the gain increases as the complexity of the target task ($\kappa = \lambda_{\min}/\lambda_{\max}$) increases ($\lambda_{\max} = 10$ and $r_* = 50$ ).

We proceed by discussing the model complexity for each of the architectures, in terms of model size. The number of parameters for ResNet is given by $2dr_*$, for GRN-v1 is given by $2dr_* + T(T-1)/2$, and for GRN-v2 is given by $2dr_* + dT(T-1)/2$. Note that by Theorem 4.2, if GRN-v1 and GRN-v2 achieve better Excess risk-model size trade-off compared to ResNet, then we can make this improvement arbitrarily strong by scaling $A - I$ (and so $\Delta$).

In the next theorem, we focus on GRN-v1 and GRN-v2 and provide sufficient conditions under which they achieve a better excess risk-model size trade-off. In the second part of the theorem, we also lower bound the improvement that GRN-v1 and GRN-v2 achieve in excess risk compared to ResNet, with using the same number of parameters.

**Theorem 4.3.** *Assume that $A - I \succeq 0$ and let $\lambda_{\max}$ and $\lambda_{\min} > 0$ respectively denote the maximum and the minimum eigenvalues of $A - I$. Define $\kappa := \lambda_{\min}/\lambda_{\max} \leq 1$. Consider a ResNet model with collective rank $r_* := \sum_{t=1}^{T} r_t$.*

*(i) If*

$$\frac{r_*}{d} \leq (1 + \kappa(\sqrt{\kappa^2 + 1} - \kappa))^2 - 1, \qquad (4.1)$$

*then GRN-v1 achieves a better excess risk-model size trade-off compared to ResNet. In addition, if*

$$r_* \leq (1 + \kappa(\sqrt{\kappa^2 + d} - \kappa))^2 - 1, \qquad (4.2)$$

*then GRN-v2 achieves a better trade-off compared to ResNet.*

*Also, GRN-v3 achieves a better trade-off compared to ResNet, if*

$$r_* \leq (1 + \eta(\sqrt{\eta^2 + d} - \eta))^2 - 1.6, \qquad (4.3)$$

*where $\eta = \sqrt{\frac{(\kappa(1+\xi_0) - \xi_0)^2 + \xi_0}{1+\xi_0}}$ and $\xi_0 = \frac{1}{\pi(d^2-1)}$.*

*(ii) Consider $\mathcal{C}_{\mathrm{GRN-v1}}$ and $\mathcal{C}_{\mathrm{GRN-v2}}$, the class of models that can be expressed by the GRN-v1 and GRN-v2 architectures with the same number of parameters as a ResNet model with $T$ layers and collective rank $r_*$. Define $G_1 := \mathsf{ER}^*(\mathcal{C}_{\mathrm{res}}) - \mathsf{ER}^*(\mathcal{C}_{\mathrm{GRN-v1}})$ and $G_2 := \mathsf{ER}^*(\mathcal{C}_{\mathrm{res}}) - \mathsf{ER}^*(\mathcal{C}_{\mathrm{GRN-v2}})$ as the reduction in the optimal excess risk achievable by these classes compared to the optimal excess risk of ResNet. We have*

$$G_1 \geq (d - r_*)\lambda_{\min}^2 - (\sqrt{d + r_*} - \sqrt{d})^2(\lambda_{\max}^2 - \lambda_{\min}^2),$$

$$G_2 \geq (d - r_*)\lambda_{\min}^2 - (\sqrt{1 + r_*} - 1)^2(\lambda_{\max}^2 - \lambda_{\min}^2),$$

$$G_3 \geq \left(d - \frac{1}{\pi(d+1)} - r_*\right)\lambda_{\min}^2 + \frac{1}{2\pi(d+1)}\lambda_{\max}^2$$
$$- (\sqrt{1.6 + r_*} - 1)^2(\lambda_{\max}^2 - \lambda_{\min}^2).$$

Our next result quantitatively shows the reduction in the collective rank one can achieve by GRNs, while maintaining the same test error as ResNet.

**Proposition 4.4.** *Consider a ResNet with collective rank $r_* = \sum_{t=1}^{T} r_t < d$. A GRN-v1 or GRN-v2 model can achieve a smaller test error with collective rank $r'_*$, where $r'_* := \frac{r_* - d\kappa^2}{1 - \kappa^2} < r_*$. Likewise, a GRN-v3 model achieve a smaller test error with collective rank $\tilde{r}_*$, where $\tilde{r}'_* := \frac{r_* - d\eta^2}{1 - \eta^2} < r_*$, with $\eta = \sqrt{\frac{(\kappa(1+\xi_0) - \xi_0)^2 + \xi_0}{1+\xi_0}}$ and $\xi_0 = \frac{1}{\pi(d^2-1)}$.*

### 4.3. Insights from the analysis

Theorem 4.3 allows us to elucidate the role of different factors on the gain achieved by GRNs.

**Role of target task complexity.** Note that $\kappa = \lambda_{\min}/\lambda_{\max} \in [0, 1]$ is a measure of complexity of the target task. Specifically, as $\kappa$ decreases, the matrix $A$ becomes closer to a low rank matrix, and hence learning it with low rank models becomes easier. Observe that the thresholds

given by the right hand side of (4.1)-(4.3) are increasing in $\kappa$, i.e., for more complex tasks we see a wider range of collective rank where GRNs outperforms the trade-off achieved by ResNet. Another way to interpret Theorem 4.3(i) is that for a fixed target task (and so fixed $\kappa$), if the collective rank $r_*$ is above this threshold, the ResNet is already rich enough that it is hard to improve upon its trade-off.

**Role of collective rank.** Observe that the lower bound on the gains $G_1$, $G_2$, $G_3$ given by Theorem 4.3(ii) are decreasing in $r_*$. In other words, when the collective rank $r_*$ of ResNet becomes smaller, the level of information dilution occurring in ResNet increases, giving GRNs a better leverage to improve model perplexity with the same number of parameters.

**Role of input dimension.** Note that the upper bounds on $r_*$ given by (4.1) to (4.3) increase with the input dimension $d$. Furthermore, the lower bounds on the gains $G_1$, $G_2$, $G_3$ given in Theorem 4.3(ii) also increase with $d$. Therefore, for larger input dimensions, we have both a wider range for $r_*$ where GRNs outperforms the trade-off achieved by ResNet, and moreover, we obtain a larger gain in reducing model error.

We refer to Figure 5 for an illustration of these trends.

### 4.4. Extension to nonlinear models

We recall the definition of Bottleneck rank from (Jacot, 2023). For a function $f : \Omega \mapsto \mathbb{R}^d$, its Bottleneck rank, denoted by $\mathrm{rank}_{BN}(f, \Omega)$ is the smallest integer $k$ such that $f$ can be factorized as $f = h \circ g$ with inner dimension $k$ (i,e, $g : \Omega \mapsto \mathbb{R}^k$ and $h : \mathbb{R}^k \mapsto \mathbb{R}^d$) It is also closely related to the Jacobian rank of a function defined as $\mathrm{rank}_{\boldsymbol{J}}(f) = \max_{\boldsymbol{x} \in \Omega} \mathrm{rank}[\boldsymbol{J}f(\boldsymbol{x})]$. In general, $\mathrm{rank}_{\boldsymbol{J}}(f) \leq \mathrm{rank}_{BN}(f)$, but for functions of the form $f = \psi \circ \boldsymbol{A} \circ \phi$ (for a linear map $\boldsymbol{A}$ and two bijections $\psi$ and $\phi$), we have $\mathrm{rank}_{\boldsymbol{J}}(f) = \mathrm{rank}_{BN}(f) = \mathrm{rank}(\boldsymbol{A})$. These two notions of rank satisfy the following properties (Jacot, 2023):

- $\mathrm{rank}(f \circ g) \leq \min\{\mathrm{rank}(f), \mathrm{rank}(g)\}$

- $\mathrm{rank}(f + g) \leq \mathrm{rank}(f) + \mathrm{rank}(g)$

**Proposition 4.5.** *Consider an MLP with* $f_t(\boldsymbol{z}) = \boldsymbol{V}_t \varphi(\boldsymbol{U}_t \boldsymbol{z})$ *with* $\boldsymbol{U}_t \in \mathbb{R}^{r_t \times d}$, $\boldsymbol{V}_t \in \mathbb{R}^{d \times r_t}$. *Denote by* $r_* := \sum_{t=1}^{T} r_t$ *the collective rank of the network. We have*

- $\mathcal{C}_{\text{base}} \subseteq \{f : \mathrm{rank}_{BN}(f) \leq \min(r_t)_{t=1}^{T}\}$.

- $\mathcal{C}_{\text{res}} \subseteq \{id + f : \mathrm{rank}_{BN}(f) \leq r_*\}$.

- $\mathcal{C}_{\text{GRN-v1}} \subseteq \{\alpha \cdot id + f : \mathrm{rank}_{BN}(f) \leq r_*\}$.

- $\mathcal{C}_{\text{GRN-v2}} \subseteq \{g : g(\boldsymbol{x}) = \boldsymbol{D}\boldsymbol{x} + f(\boldsymbol{x}) : \mathrm{rank}_{BN}(f) \leq r_*, \boldsymbol{D} \text{ is diagonal}\}$.

## 5. Experiments

We conduct experiments on language modeling and image classification tasks to evaluate the effectiveness of DCA and to validate our theoretical insights. For the language modeling tasks, the performance of DCA is compared against the standard transformer (Vaswani, 2017) on the LM1B (Chelba et al., 2013) and C4 (Raffel et al., 2020a) datasets. Unless stated otherwise, each model has an embedding dimension of 512 and an MLP dimension of four times the embedding dimension. By default, DCA uses a stack of all the previous layer outputs as input to the GRNs. When DCA includes only the first and last-$k$ layer outputs explicitly in the input stack (see Section 3.1), then this is denoted as $k$-DCA.

Each model is trained with a sequence length of 128 and a batch size of 2048 over 64 TPUs for 500k steps, totaling 131B tokens. We use the AdamW optimizer (Loshchilov & Hutter, 2017) with $\beta_1 = 0.9$, $\beta_2 = 0.98$, a weight decay of 0.1, and a learning rate of 0.0016 with 1000 warmup steps and an inverse square root schedule (Raffel et al., 2020b).

**Model depth scaling.** For the first experiment, we pre-train a transformer and DCA on LM1B. We increase the model depth from 6 to 42 layers and show the relation between perplexity (Jelinek et al., 1977) and model size in Figure 6. The figure shows that DCA obtains a lower perplexity for a given parameter budget. Notably, the 30-layer DCA model obtains a better perplexity than the 42-layer transformer, making DCA more parameter-efficient than adding layers.

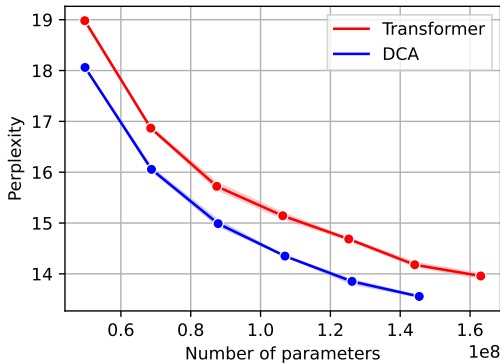

*Figure 6.* Perplexity on LM1B with 6, 12, 18, 24, 30, 36, and 42 layer transformer and DCA models.

**First and last-$k$.** DCA can be made more efficient by including only the first and last-$k$ layer outputs explicitly in the input stack to the GRNs (see Section 3.1). In this experiment, we study the effect of $k$ on a 24-layer model's efficiency and quality. Table 1 shows that reducing $k$ speeds up training while only slightly increasing the perplexity. Either small or large $k$ obtain good training efficiency, as DCA then obtains the final perplexity of the transformer in a third of the time. Setting $k = 2$ results in a model with 48%

lower inference latency compared to $k = 24$, thus setting $k$ to be small results in efficient training and fast inference.

*Table 1.* Training speed in batches per second, normalized time for a method to reach the perplexity of the transformer, and the final perplexity (PPL) of the transformer and DCA with varying $k$.

| METHOD | SPEED | TIME | PPL |
|---|---|---|---|
| TRANSFORMER | **8.14±0.18** | 1.00 | 15.14±0.06 |
| 1-DCA | 5.62±0.04 | **0.33** | 14.48±0.05 |
| 2-DCA | 5.39±0.06 | **0.33** | 14.41±0.04 |
| 4-DCA | 5.01±0.12 | 0.37 | 14.50±0.03 |
| 8-DCA | 4.35±0.14 | 0.47 | 14.49±0.02 |
| 16-DCA | 3.86±0.08 | 0.40 | 14.35±0.07 |
| 24-DCA | 3.72±0.08 | 0.39 | **14.35±0.00** |

**Training time.** The effectiveness of a model architecture heavily depends on its training efficiency. Figure 7 shows the training time-perplexity trade-off for 24, 36, and 42 layer transformer and 2-DCA models. The figure shows that 2-DCA achieves better perplexity for a given training time, highlighting the training efficiency of DCA. The training time versus perplexity results when DCA uses all previous layer outputs in the GRNs are provided in Appendix F.

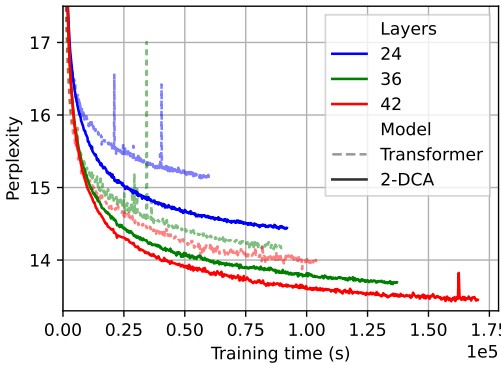

*Figure 7.* Perplexity on LM1B versus the training time with transformer and 2-DCA models of various depths.

**Model width scaling.** Our theoretical results indicate that the benefit of GRN is inversely related to the rank of the model. With this experiment, we validate whether the theoretical results carry over to the transformer architecture by varying the model width. Table 2 shows the final perplexity of a 12-layer model with an embedding dimension ranging from 64 till 1024, pre-trained on LM1B. The delta column, with the difference between the transformer and DCA, shows that the benefit of DCA is reduced as the width of the model increases, which is consistent with our theoretical results. These results are in contrast with the depth scaling results, where the improvement of DCA is maintained for deeper models.

**Model scaling.** For this experiment, we train transformer

*Table 2.* Perplexity on LM1B for models of varying widths.

| WIDTH | TRANSFORMER | DCA | DELTA |
|---|---|---|---|
| 64 | 45.75±0.06 | 42.94±0.07 | -2.82 |
| 192 | 25.49±0.15 | 23.92±0.04 | -1.57 |
| 384 | 18.86±0.04 | 17.83±0.04 | -1.03 |
| 768 | 14.70±0.04 | 14.11±0.07 | -0.59 |
| 1024 | 13.61±0.01 | 13.22±0.06 | -0.39 |

and 8-DCA models of increasing size on the C4 dataset. The results in Table 3 show that DCA consistently outperforms the standard transformer model. The absolute improvement in perplexity decreases for large models, which is consistent with the width scaling results. The perplexity throughout training is provided in Appendix G.

*Table 3.* Perplexity on C4 for models of varying depths and widths.

| D | W | PARAMS | TRANSF. | 8-DCA | DELTA |
|---|---|---|---|---|---|
| 9 | 771 | 75M | 27.876 | 26.443 | -1.443 |
| 18 | 771 | 124M | 23.013 | 21.810 | -1.203 |
| 13 | 1111 | 179M | 21.570 | 20.461 | -1.109 |
| 18 | 1111 | 234M | 19.756 | 18.824 | -0.932 |
| 18 | 1600 | 449M | 17.166 | 16.764 | -0.402 |

**Retrofitting pre-trained models.** Since our method is identical to a standard residual network at initialization, adding DCA to a pre-trained model does not alter its function. In Table 4, we compare continuing training the pre-trained model with adding DCA to the pre-trained model. Incorporating DCA results in a perplexity improvement of 0.19 after 60k extra training steps, compared to just 0.02 for the transformer. Thus, pre-trained models with a residual architecture can also benefit from incorporating DCA.

*Table 4.* Perplexity on LM1B for extended training of 6-layer models. DCA is added to a 500k steps pre-trained transformer.

| STEPS | TRANSFORMER | DCA | DELTA |
|---|---|---|---|
| 500K | 18.98±0.01 | 18.98±0.01 | 0.00 |
| 500K + 20K | 18.96±0.02 | 18.81±0.01 | -0.15 |
| 500K + 40K | 18.96±0.01 | 18.79±0.03 | -0.17 |
| 500K + 60K | 18.96±0.01 | 18.79±0.04 | -0.17 |

**Training stability.** The occurrence of loss spikes is a problem when training large models as they can disrupt an expensive training run (Chowdhery et al., 2023). In Figures 7 and 8, we indeed observe clear loss spikes with the transformer model. Interestingly, training DCA is more stable, showing no significant loss spikes even for large models. This constitutes an important benefit of DCA.

**Comparison with related work.** We compare the perplexity of DCA with those obtained by the recent related works LAuReL (Menghani et al., 2024), DenseFormer (Pagliardini

et al., 2024), and Hyper-Connections (dynamic) (Zhu et al., 2024) in Table 5. DCA improves upon the prior best method, hyper-connections, with a difference in perplexity of 0.59, which is the biggest improvement among the methods.

*Table 5.* Perplexity (PPL) and parameter count on LM1B using a 6-layer model, comparing DCA with related work.

| METHOD | PARAMS | PPL |
|---|---|---|
| TRANSFORMER | 49.65M | 18.98±0.01 |
| LAURELL-PA | 49.75M | 18.99±0.05 |
| 1X1-DENSEFORMER | 49.65M | 18.80±0.11 |
| HYPER-CONNECTIONS | 49.68M | 18.65±0.03 |
| DCA (OURS) | 49.73M | **18.06±0.01** |

*Table 6.* Perplexity (PPL) on C4 using a 13-layer model and embedding dimension 1111, comparing DCA with related work. The baseline model has roughly 179M parameters.

| METHOD | PPL |
|---|---|
| TRANSFORMER | 21.534 |
| LAURELL-PA | 20.951 |
| 1X1-DENSEFORMER | 21.168 |
| HYPER-CONNECTIONS (STACK SIZE=4) | 21.077 |
| HYPER-CONNECTIONS (STACK SIZE=10) | 20.718 |
| 8-DCA (OURS) | **20.392** |

**Ablation study.** To determine the relative gain of each of the proposed generalizations, in Table 7 we show the perplexity obtained by each method described in Section 3. The GRN versions use one GRN instance per decoder block. DCA, in contrast, uses three independent instances of GRN-v3 per decoder block. The biggest improvement in perplexity comes from GRN-v1, followed by DCA and GRN-v2.

*Table 7.* Ablation study of DCA, showing the parameter count and the perplexity (PPL) on LM1B with a 6-layer model.

| ABLATION | PARAMS | PPL |
|---|---|---|
| TRANSFORMER | 49.65M | 18.98±0.02 |
| GRN-v1 | 49.65M | 18.80±0.11 |
| GRN-v2 | 49.66M | 18.43±0.04 |
| GRN-v3 | 49.68M | 18.41±0.10 |
| DCA | 49.73M | **18.06±0.01** |

**ImageNet classification.** In addition to the language modelling experiments, we also experiment with image classification using the ImageNet dataset and the vision transformer (ViT) model (Dosovitskiy et al., 2021). Since the ViT model is transformer-based, DCA can be incorporate in the same way as for the language models presented earlier. In Table 8, we present the results on the ViT-S/16 model (22M parameters) and follow the experimental setup by Beyer et al. (2022). The results show a 0.7% improvement in

classification accuracy, demonstrating that DCA effectively generalizes to the vision domain.

*Table 8.* Loss and Accuracy on ImageNet classification.

| METHOD | LOSS | ACCURACY |
|---|---|---|
| ViT | 0.5698 | 76.4 |
| ViT + DCA (OURS) | **0.5284** | **77.1** |

## 6. Conclusion

This paper introduces DeepCrossAttention (DCA), a novel transformer architecture that enhances the flow of information across layers. It achieves lower perplexity for a given parameter budget and training time for a minimal increase in model parameters. DCA enables dynamic interactions between layer outputs by building on three generalizations of the standard residual network (GRN). We showed theoretically that GRN obtains a better test error-model complexity trade-off. In our DCA experiments we observe significant improvements in model stability, convergence, and quality.

## Acknowledgment

Adel Javanmard is supported in part by the NSF Award DMS-2311024, the Sloan fellowship in Mathematics, an Adobe Faculty Research Award, an Amazon Faculty Research Award, and an iORB grant from USC Marshall School of Business. The authors are grateful to anonymous reviewers for their feedback on improving this paper.

## Impact Statement

This paper presents work whose goal is to advance the field of Machine Learning. There are many potential societal consequences of our work, none which we feel must be specifically highlighted here.

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

# A. Proof of Theorem 4.1

We restate each of the claims in the theorem statement, followed by its proof.

- $\mathcal{C}_{\text{base}} = \{ \boldsymbol{x} \mapsto \boldsymbol{M} \boldsymbol{x} : \text{rank}(\boldsymbol{M}) \leq \min(r_t)_{t=1}^T \}$.

Note that by the inequality $\text{rank}(\boldsymbol{A}\boldsymbol{B}) \leq \min\{\text{rank}(\boldsymbol{A}), \text{rank}(\boldsymbol{B})\}$, if $\boldsymbol{M}$ is of the form $\prod_{t=1}^T \boldsymbol{V}_t$ then $\text{rank}(\boldsymbol{M}) \leq \min(r_t)_{t=1}^T$. For the other direction consider any matrix $\boldsymbol{M}$ with $\text{rank}(\boldsymbol{M}) = r_0 \leq \min(r_t)_{t=1}^T$, and its SVD as $\boldsymbol{M} = \boldsymbol{P}\boldsymbol{S}\boldsymbol{Q}^\mathsf{T}$ with $\boldsymbol{P}, \boldsymbol{Q} \in \mathbb{R}^{d \times r_0}$ with full column ranks. By setting, $\boldsymbol{V}_1 = \boldsymbol{P}\boldsymbol{S}\boldsymbol{Q}^\mathsf{T}$ and $\boldsymbol{V}_2 = \ldots = \boldsymbol{V}_T = \boldsymbol{Q}\boldsymbol{Q}^\mathsf{T}$ we have $\boldsymbol{M} = \prod_{t=1}^T \boldsymbol{V}_t$, because $\boldsymbol{Q}^\mathsf{T}\boldsymbol{Q} = \boldsymbol{I}$ and also $\text{rank}(\boldsymbol{V}_t) = r_0 \leq \min(r_t)_{t=1}^T \leq r_t$.

- $\mathcal{C}_{\text{res}} = \{ \boldsymbol{x} \mapsto (\boldsymbol{I} + \boldsymbol{M}) \boldsymbol{x} : \text{rank}(\boldsymbol{M}) \leq r_* \}$

We have

$$\prod_{t=1}^T (\boldsymbol{I} + \boldsymbol{V}_t) = \boldsymbol{V}_1 \prod_{t=2}^T (\boldsymbol{I} + \boldsymbol{V}_t) + \prod_{t=2}^T (\boldsymbol{I} + \boldsymbol{V}_t)$$

$$= \boldsymbol{V}_1 \prod_{t=2}^T (\boldsymbol{I} + \boldsymbol{V}_t) + \boldsymbol{V}_2 \prod_{t=3}^T (\boldsymbol{I} + \boldsymbol{V}_t) + \prod_{t=3}^T (\boldsymbol{I} + \boldsymbol{V}_t)$$

$$= \ldots$$

$$= \boldsymbol{I} + \boldsymbol{V}_T + \sum_{t=1}^{T-1} \boldsymbol{V}_t \prod_{\tau=t+1}^T (\boldsymbol{I} + \boldsymbol{V}_\tau)$$

Note that each of the summand is of rank at most $r_t$, so it can be written as $\boldsymbol{I} + \boldsymbol{M}$ with $\text{rank}(\boldsymbol{M}) \leq \sum_{t=1}^T r_t$. Hence $\prod_{t=1}^T (\boldsymbol{I} + \boldsymbol{V}_t) \boldsymbol{x} \in \mathcal{C}_{\text{res}}$.

We next show that any $\boldsymbol{I} + \boldsymbol{M}$ with $\text{rank}(\boldsymbol{M}) := r \leq \sum_{t=1}^T r_t$ can be written as $\prod_{t=1}^T (\boldsymbol{I} + \boldsymbol{V}_t)$ with $\text{rank}(\boldsymbol{V}_t) \leq r_t$ for $t \in [T]$. We show this claim by induction. For the basis ($T = 1$), we can take $\boldsymbol{V}_1 = \boldsymbol{M}$. To complete the induction step, we need to find $\boldsymbol{V} \in \mathbb{R}^{d \times d}$ such that $\text{rank}(\boldsymbol{V}) = r_T$ and $(\boldsymbol{I} + \boldsymbol{V})^{-1}(\boldsymbol{I} + \boldsymbol{M}) - \boldsymbol{I}$ is of rank at most $\sum_{t=1}^{T-1} r_t$. Then by the induction hypothesis, we can write

$$(\boldsymbol{I} + \boldsymbol{V})^{-1}(\boldsymbol{I} + \boldsymbol{M}) = \prod_{t=1}^{T-1} (\boldsymbol{I} + \boldsymbol{V}_t),$$

with $\text{rank}(\boldsymbol{V}_t) \leq r_t$, which completes the proof. Without loss of generality, we assume $r_T \leq r$; otherwise we can take $\boldsymbol{V}_T = \boldsymbol{M}$ and $\boldsymbol{V}_t = \boldsymbol{0}$ for $t \leq T - 1$.

To find such $\boldsymbol{V}$ we write $\boldsymbol{M} = \boldsymbol{P}\boldsymbol{Q}^\mathsf{T}$ with $\boldsymbol{P}, \boldsymbol{Q} \in \mathbb{R}^{d \times r}$ having full column rank. Define $\boldsymbol{P}_1, \boldsymbol{Q}_1 \in \mathbb{R}^{d \times r_T}$ obtaining by considering the first $r_T$ columns of $\boldsymbol{P}$ and $\boldsymbol{Q}$. Additionally, define

$$\boldsymbol{B} := \boldsymbol{P}_1 (\boldsymbol{I} + \boldsymbol{Q}_1^\mathsf{T} \boldsymbol{P}_1)^{-1}, \quad \boldsymbol{C} = \boldsymbol{Q}_1 (\boldsymbol{I} + \boldsymbol{P}_1^\mathsf{T} \boldsymbol{Q}_1). \tag{A.1}$$

We next construct $\boldsymbol{V}$ by setting $\boldsymbol{V} := \boldsymbol{B}\boldsymbol{C}^\mathsf{T}$. Clearly, $\text{rank}(\boldsymbol{V}) = r_T$. We also have

$$(\boldsymbol{I} + \boldsymbol{V})^{-1}(\boldsymbol{I} + \boldsymbol{M}) - \boldsymbol{I} = (\boldsymbol{I} + \boldsymbol{B}\boldsymbol{C}^\mathsf{T})^{-1}\boldsymbol{M} + (\boldsymbol{I} + \boldsymbol{B}\boldsymbol{C}^\mathsf{T})^{-1} - \boldsymbol{I}$$

$$= (\boldsymbol{I} + \boldsymbol{B}\boldsymbol{C}^\mathsf{T})^{-1}\boldsymbol{M} + \boldsymbol{I} - \boldsymbol{B}(\boldsymbol{I} + \boldsymbol{C}^\mathsf{T}\boldsymbol{B})^{-1}\boldsymbol{C}^\mathsf{T} - \boldsymbol{I}$$

$$= (\boldsymbol{I} + \boldsymbol{B}\boldsymbol{C}^\mathsf{T})^{-1}(\boldsymbol{P}_1\boldsymbol{Q}_1^\mathsf{T} + \boldsymbol{P}_{\sim 1}\boldsymbol{Q}_{\sim 1}^\mathsf{T}) - \boldsymbol{B}(\boldsymbol{I} + \boldsymbol{C}^\mathsf{T}\boldsymbol{B})^{-1}\boldsymbol{C}^\mathsf{T}.$$

Here we consider the notation $\boldsymbol{P} = [\boldsymbol{P}_1 | \boldsymbol{P}_{\sim 1}]$ and $\boldsymbol{Q} = [\boldsymbol{Q}_1 | \boldsymbol{Q}_{\sim 1}]$. The second step above follows from the Woodbury matrix identity. Rearranging the terms we have

$$(\boldsymbol{I} + \boldsymbol{V})^{-1}(\boldsymbol{I} + \boldsymbol{M}) - \boldsymbol{I} = (\boldsymbol{I} + \boldsymbol{B}\boldsymbol{C}^\mathsf{T})^{-1}\boldsymbol{P}_{\sim 1}\boldsymbol{Q}_{\sim 1}^\mathsf{T} + (\boldsymbol{I} + \boldsymbol{B}\boldsymbol{C}^\mathsf{T})^{-1}\boldsymbol{P}_1\boldsymbol{Q}_1^\mathsf{T} - \boldsymbol{B}(\boldsymbol{I} + \boldsymbol{C}^\mathsf{T}\boldsymbol{B})^{-1}\boldsymbol{C}^\mathsf{T}. \tag{A.2}$$

The first term above is of rank at most $\text{rank}(\boldsymbol{P}_{\sim 1}) = r - r_T \leq \sum_{t=1}^{T-1} r_t$. We next show that the second and the third term cancel each other. Equivalently, we show that

$$\boldsymbol{P}_1\boldsymbol{Q}_1^\mathsf{T} = (\boldsymbol{I} + \boldsymbol{B}\boldsymbol{C}^\mathsf{T})\boldsymbol{B}(\boldsymbol{I} + \boldsymbol{C}^\mathsf{T}\boldsymbol{B})^{-1}\boldsymbol{C}^\mathsf{T}. \tag{A.3}$$

To do this, we next show that

$$P_1 = (I + BC^\mathsf{T})B, \quad Q_1^\mathsf{T} = (I + C^\mathsf{T}B)^{-1}C^\mathsf{T}. \tag{A.4}$$

Recalling (A.1) we have $P_1 = B(I + Q_1^\mathsf{T}P_1)$. Also

$$
\begin{aligned}
C^\mathsf{T}B &= (I + Q_1^\mathsf{T}P_1)Q_1^\mathsf{T}P_1(I + Q_1^\mathsf{T}P_1)^{-1} \\
&= (I + Q_1^\mathsf{T}P_1)(I + Q_1^\mathsf{T}P_1 - I)(I + Q_1^\mathsf{T}P_1)^{-1} \\
&= (I + Q_1^\mathsf{T}P_1)(I - (I + Q_1^\mathsf{T}P_1)^{-1}) \\
&= I + Q_1^\mathsf{T}P_1 - I \\
&= Q_1^\mathsf{T}P_1
\end{aligned}
\tag{A.5}
$$
$$\tag{A.6}$$

Therefore, $P_1 = B(I + C^\mathsf{T}B) = (I + BC^\mathsf{T})B$. Likewise, recalling (A.1) we have $Q_1 = C(I + P_1^\mathsf{T}Q_1)^{-1}$. Hence,

$$Q_1^\mathsf{T} = (I + Q_1^\mathsf{T}P_1)^{-1}C^\mathsf{T} = (I + C^\mathsf{T}B)^{-1}C^\mathsf{T},$$

using (A.6). This completes the proof of (A.4) and so (A.3).

Invoking (A.2) we get

$$(I + V)^{-1}(I + M) - I = (I + BC^\mathsf{T})^{-1}P_{\sim 1}Q_{\sim 1}^\mathsf{T},$$

which is of rank at most $r - r_T \le \sum_{t=1}^{T-1} r_t$, which completes the proof of the induction step.

- $\mathcal{C}_{\mathrm{GRN-v1}} = \{x \mapsto (\alpha I + M)x : \ \mathrm{rank}(M) \le r_*\}$.

We prove this claim by induction. The induction basis $(T = 0)$ follows readily since $G_1 b_1 = b_1 x$. Assume the induction hypothesis for $t$. We have $f_t(g_t(x)) = V_t G_t b_t$ and so $G_{t+1} = [V_t G_t b_t \mid G_t]$. Writing $b_{t+1} = \begin{bmatrix} b_1 \\ b_{\sim 1} \end{bmatrix}$ we obtain

$$G_{t+1}b_{t+1} = V_t G_t b_t b_1 + G_t b_{\sim 1}.$$

By induction hypothesis, $G_t b_{\sim 1}$ is the set of functions of the form $(\alpha I + M)x$ with $\mathrm{rank}(M) \le \sum_{\ell=1}^{t-1} r_\ell$.

Since $\mathrm{rank}(V_t) \le r_t$ the set of functions that can be represented as $G_{t+1}b_{t+1}$ is a subset of $(\alpha I + M)x$ with $\mathrm{rank}(M) \le \sum_{\ell=1}^{t} r_\ell$. Conversely, any given $M$ of rank $\sum_{\ell=1}^{t} r_\ell$ can be written as $M = M_1 + V$ with $\mathrm{rank}(M_1) = \sum_{\ell=1}^{t-1} r_\ell$ and $\mathrm{rank}(V) = r_t$. By induction hypothesis, $(\alpha I + M_1)x$ can be expressed by the term $G_t b_{\sim 1}$. In addition, $Vx$ can also be expressed by the term $V_t G_t b_t b_1$, by taking $V_t = V$, $b_t = (0, 0, \ldots, 1)^\mathsf{T}$, $b_1 = 1$, which is possible since they are free from the choice of $b_{\sim 1}$.

Hence, $G_{t+1}b_{t+1}$ the set of functions of the form $(\alpha I + M)x$ with $\mathrm{rank}(M) \le \sum_{\ell=1}^{t} r_\ell$, completing the induction step.

- $\mathcal{C}_{\mathrm{GRN-v2}} = \{x \mapsto (D + M)x : \ \mathrm{rank}(M) \le r_*, D \text{ is diagonal}\}$.

The proof follows similar to that of GRN-v1. For the induction basis $(T = 0)$, we have $(G_1 \odot b_1)\mathbf{1} = (x \odot b_1)\mathbf{1} = \mathrm{diag}(b_1)x$. Assume the induction hypothesis for $t$. We have $f_t(g_t(x)) = V_t(G_t \odot b_t)\mathbf{1}$ and so $G_{t+1} = [V_t(G_t \odot b_t)\mathbf{1} \mid G_t]$. Writing $b_{t+1} = \left[b_{t+1}^{(1)} | b_{t+1}^{(\sim 1)}\right]$ we obtain

$$G_{t+1} \odot b_{t+1} = \mathrm{diag}(b_{t+1}^{(1)})V_t(G_t \odot b_t)\mathbf{1} + G_t \odot b_{t+1}^{(\sim 1)},$$

and hence

$$(G_{t+1} \odot b_{t+1})\mathbf{1} = \mathrm{diag}(b_{t+1}^{(1)})V_t(G_t \odot b_t)\mathbf{1} + (G_t \odot b_{t+1}^{(\sim 1)})\mathbf{1}.$$

By induction hypothesis, $(G_t \odot b_{t+1}^{(\sim 1)})\mathbf{1}$ is the set of functions of the form $(D + M)x$ with $\mathrm{rank}(M) \le \sum_{\ell=1}^{t-1} r_\ell$. By varying $V_t, b_t$ and $b_t^{(1)}$, the term $\mathrm{diag}(b_{t+1}^{(1)})V_t(G_t \odot b_t)\mathbf{1}$ covers all functions of the form $Vx$ with $V$ a $d \times d$ matrices of rank $r_t$ (for given $V$ of rank $r_t$, take $V_t = V$, $b_t = [0|0| \ldots |\mathbf{1}]$, $b_{t+1}^{(1)} = \mathbf{1}$, which is possible since they are free from

the choice of $b_{t+1}^{(\sim 1)}$). Hence, $(G_{t+1} \odot b_{t+1})\mathbf{1}$ is the set of functions of the form $(D + M)x$ with $\mathrm{rank}(M) \leq \sum_{\ell=1}^{t} r_\ell$, completing the induction step.

- $\mathcal{C}_{\mathrm{GRN-v3}} \supset \{x \mapsto (D + M)x + \sigma(w^\mathsf{T}x)x : \ \mathrm{rank}(M) \leq r_*, D \text{ is diagonal}, w \in \mathbb{R}^{d \times 1}\}$.

We prove that

$$\mathcal{C}_{\mathrm{GRN-v3}} \supset \left\{ x \mapsto \sum_{t=1}^{T} V_t x + \sigma(w^\mathsf{T}x)x + Dx, \ \mathrm{rank}(V_t) = r_t, \ D \text{ is diagonal} \right\}.$$

we show the claim by induction on the number of layers. For the basis $T = 0$, we have

$$(G_1 \odot (b_1 + \bar{w}_1))\mathbf{1} = \mathrm{diag}(b_1 + \bar{w}_1)x = \mathrm{diag}(b_1)x + \sigma(w_1^\mathsf{T}x)x.$$

Assume the induction hypothesis for $t$. We have $f_t(g_t(x)) = V_t(G_t \odot c_t)\mathbf{1}$ with $c_t := b_t + \mathbf{1}\sigma(w_t^\mathsf{T}G_t)$ and so $G_{t+1} = [V_t(G_t \odot c_t)\mathbf{1} \mid G_t]$. Writing $c_{t+1} = \left[c_{t+1}^{(1)} \mid c_{t+1}^{(\sim 1)}\right]$ we obtain

$$G_{t+1} \odot c_{t+1} = \mathrm{diag}(c_{t+1}^{(1)})V_t(G_t \odot c_t)\mathbf{1} + G_t \odot c_{t+1}^{(\sim 1)},$$

and hence

$$(G_{t+1} \odot c_{t+1})\mathbf{1} = \mathrm{diag}(c_{t+1}^{(1)})V_t(G_t \odot c_t)\mathbf{1} + (G_t \odot c_{t+1}^{(\sim 1)})\mathbf{1}.$$

We take $b_t = [\mathbf{0}|\mathbf{0}|\dots|\mathbf{1}]$ and $w_t = \mathbf{0}$, and so $c_t = [\mathbf{0}|\mathbf{0}|\dots|\mathbf{1}]$. We also take $c_{t+1}^{(1)} = \mathbf{1}$. So,

$$(G_{t+1} \odot c_{t+1})\mathbf{1} = V_t x + (G_t \odot c_{t+1}^{(\sim 1)})\mathbf{1}.$$

By induction hypothesis, $(G_t \odot c_{t+1}^{(\sim 1)})\mathbf{1}$ covers all the functions of the form $x \mapsto \sum_{\ell=1}^{t-1} V_\ell x + \sigma(w^\mathsf{T}x)x + Dx$, which along with the above equation completes the induction step.

Finally, note that any matrix $M \in \mathbb{R}^{d \times d}$ of rank $r_* = \sum_{t=1}^{T} r_t$ can be written as $\sum_{t=1}^{T} V_t$, for some choices of matrices $V_t \in \mathbb{R}^{d \times d}$ of rank $r_t$.

## B. Proof of Theorem 4.2

By Eckart–Young–Mirsky theorem, $U_{r*}\Sigma_{r*}V_{r*}^\mathsf{T}$ is the best rank $r*$ approximation to $A - I$, by which we obtain $\mathsf{ER}^*(\mathcal{C}_{\mathrm{res}}) = \|\Delta\|_F^2$.

We also have by definition,

$$\mathsf{ER}^*(\mathcal{C}_{\mathrm{GRN-v1}}) = \min_{\alpha, \mathrm{rank}(\tilde{A})=r_*} \left\| A - \alpha I - \tilde{A} \right\|_F^2. \tag{B.1}$$

Recall the SVD of $A - I = U\Sigma V^\mathsf{T}$ and consider the following decompositions:

$$U = [U_{r_*} \mid U_{r_*,\perp}], \quad V = [V_{r_*} \mid V_{r_*,\perp}], \quad \Sigma = \begin{bmatrix} \Sigma_{r_*} & \mathbf{0} \\ \mathbf{0} & \Sigma_{r_*,\perp} \end{bmatrix},$$

with $U_{r_*,\perp}, V_{r_*,\perp} \in \mathbb{R}^{d \times (d-r_*)}$, and $\Sigma_{r_*,\perp}$ a diagonal matrix of size $d - r_*$. Since $U$ is unitary matrix, we have $U_{r_*}U_{r_*}^\mathsf{T} + U_{r_*,\perp}U_{r_*,\perp}^\mathsf{T} = I$.

We then note that for any choice of $\alpha$, $\tilde{A}$, we have

$$\begin{aligned}
A - \alpha I - \tilde{A} &= A - I + (1 - \alpha)I - \tilde{A} \\
&= \Delta + U_{r*}\Sigma_{r*}V_{r*}^\mathsf{T} + (1 - \alpha)I - \tilde{A} \\
&= \Delta + U_{r*}\Sigma_{r*}V_{r*}^\mathsf{T} + (1 - \alpha)U_{r*}U_{r*}^\mathsf{T} + (1 - \alpha)U_{r_*,\perp}U_{r_*,\perp}^\mathsf{T} - \tilde{A}.
\end{aligned}$$

Next, by taking $\tilde{A} = U_{r*}\Sigma_{r*}V_{r*}^{\mathsf{T}} + (1-\alpha)U_{r*}U_{r*}^{\mathsf{T}} = U_{r*}[\Sigma_{r*}V_{r*}^{\mathsf{T}} + (1-\alpha)U_{r*}^{\mathsf{T}}]$ as the rank-$r_*$ matrix, we obtain

$$A - \alpha I - \tilde{A} = \Delta + (1-\alpha)U_{r_*,\perp}U_{r_*,\perp}^{\mathsf{T}}.$$

Invoking the characterization (B.1), we arrive at

$$\begin{aligned}
\mathsf{ER}^*(\mathcal{C}_{\mathrm{GRN-v1}}) &\leq \min_{\alpha} \left\| \Delta + (1-\alpha)U_{r_*,\perp}U_{r_*,\perp}^{\mathsf{T}} \right\|_F^2 \\
&= \min_{\tilde{\alpha}} \|\Delta\|_F^2 + \tilde{\alpha}^2 \left\| U_{r_*,\perp}U_{r_*,\perp}^{\mathsf{T}} \right\|_F^2 - 2\tilde{\alpha}\,\mathrm{trace}(\Delta^{\mathsf{T}}U_{r_*,\perp}U_{r_*,\perp}^{\mathsf{T}}) \\
&= \min_{\tilde{\alpha}} \|\Delta\|_F^2 + (d-r_*)\tilde{\alpha}^2 - 2\tilde{\alpha}\,\mathrm{trace}(\Delta) \\
&= \|\Delta\|_F^2 - \frac{1}{d-r_*}\mathrm{trace}(\Delta)^2,
\end{aligned}$$

where in the second equality, we used the fact that $U_{r_*,\perp}$ is unitary and so $\left\| U_{r_*,\perp}U_{r_*,\perp}^{\mathsf{T}} \right\|_F^2 = d - r_*$. In addition, observed that

$$\Delta = A - I - U_{r_*}\Sigma_{r_*}V_{r_*}^{\mathsf{T}} = U\Sigma V^{\mathsf{T}} - U_{r_*}\Sigma_{r_*}V_{r_*}^{\mathsf{T}} = U_{r_*,\perp}\Sigma_{r_*,\perp}V_{r_*,\perp}^{\mathsf{T}}.$$

Therefore,

$$\Delta^{\mathsf{T}}U_{r_*,\perp}U_{r_*,\perp}^{\mathsf{T}} = V_{r_*,\perp}\Sigma_{r_*,\perp}U_{r_*,\perp}^{\mathsf{T}}U_{r_*,\perp}U_{r_*,\perp}^{\mathsf{T}} = V_{r_*,\perp}\Sigma_{r_*,\perp}U_{r_*,\perp}^{\mathsf{T}} = \Delta^{\mathsf{T}},$$

and so $\mathrm{trace}(\Delta^{\mathsf{T}}U_{r_*,\perp}U_{r_*,\perp}^{\mathsf{T}}) = \mathrm{trace}(\Delta^{\mathsf{T}}) = \mathrm{trace}(\Delta)$. This completes the proof of the upper bound on $\mathsf{ER}^*(\mathcal{C}_{\mathrm{GRN-v1}})$.

For $\mathsf{ER}^*(\mathcal{C}_{\mathrm{GRN-v2}})$ we have

$$\begin{aligned}
\mathsf{ER}^*(\mathcal{C}_{\mathrm{GRN-v2}}) &\leq \min_{D \text{ diagonal}} \left\| A - D - U_{r*}\Sigma_{r*}V_{r*}^{\mathsf{T}} \right\|_F \\
&= \min_{D \text{ diagonal}} \left\| \Delta + I - D \right\|_F^2 \\
&= \min_{\tilde{D} \text{ diagonal}} \left\| \Delta - \tilde{D} \right\|_F^2 \\
&= \|\Delta\|_F^2 - \sum_{i=1}^{d} \Delta_{ii}^2.
\end{aligned} \tag{B.2}$$

In addition, since GRN-v2 optimizes over a larger class of models (using diagonals instead of scale of identity), we have

$$\mathsf{ER}^*(\mathcal{C}_{\mathrm{GRN-v2}}) \leq \mathsf{ER}^*(\mathcal{C}_{\mathrm{GRN-v1}}) \leq \|\Delta\|_F^2 - \frac{1}{d-r_*}\mathrm{trace}(\Delta)^2 \tag{B.3}$$

Combining (B.2) and (B.3) we obtain the claimed upper bound on $\mathsf{ER}^*(\mathcal{C}_{\mathrm{GRN-v2}})$.

We next proceed to bound $\mathsf{ER}^*(\mathcal{C}_{\mathrm{GRN-v3}})$. By Theorem 4.1, this quantity is at most the optimal objective value of the following optimization problem:

$$\min_{D,M,w} \mathbb{E}\left[ \left\| Ax - Dx - Mx - \sigma(w^{\mathsf{T}}x)x \right\|_{\ell_2}^2 \right] \tag{B.4}$$
$$\text{subject to} \quad \mathrm{rank}(M) \leq r_*, \; D \text{ is diagonal}, \; w \in \mathbb{R}^d.$$

We calculate the expectation in the objective over the gaussian vector $x \sim \mathsf{N}(0, I)$. Define the shorthand $B := A - D - M$. We write

$$\begin{aligned}
\mathbb{E}\left[ \left\| Bx - \sigma(w^{\mathsf{T}}x)x \right\|_{\ell_2}^2 \right] &= \|B\|_F^2 + \mathbb{E}[\sigma(w^{\mathsf{T}}x)^2 \|x\|_{\ell_2}^2 - 2\sigma(w^{\mathsf{T}}x)x^{\mathsf{T}}Bx] \\
&= \|B\|_F^2 + \mathbb{E}[\sigma(w^{\mathsf{T}}x)^2 \|x\|_{\ell_2}^2] - 2\mathrm{trace}\left( B\,\mathbb{E}[\sigma(w^{\mathsf{T}}x)xx^{\mathsf{T}}] \right)
\end{aligned} \tag{B.5}$$

These two expectations are characterized by our next lemma.

**Lemma B.1.** *Suppose that $\boldsymbol{x} \sim \mathsf{N}(0, \boldsymbol{I})$ and let $\sigma(z) = z\boldsymbol{1}(z \geq 0)$ be the ReLu function. Then,*

$$\mathbb{E}[\sigma(\boldsymbol{w}^\mathsf{T}\boldsymbol{x})^2 \|\boldsymbol{x}\|_{\ell_2}^2] = \|\boldsymbol{w}\|_{\ell_2}^2 \frac{d+1}{2}, \tag{B.6}$$

$$\mathbb{E}[\sigma(\boldsymbol{w}^\mathsf{T}\boldsymbol{x})\boldsymbol{x}\boldsymbol{x}^\mathsf{T}] = \frac{1}{\sqrt{2\pi}}\left(\|\boldsymbol{w}\|_{\ell_2} \boldsymbol{I} + \frac{\boldsymbol{w}\boldsymbol{w}^\mathsf{T}}{\|\boldsymbol{w}\|_{\ell_2}}\right). \tag{B.7}$$

Using the result of Lemma B.1 in (B.5) we obtain

$$\mathbb{E}\left[\|\boldsymbol{B}\boldsymbol{x} - \sigma(\boldsymbol{w}^\mathsf{T}\boldsymbol{x})\boldsymbol{x}\|_{\ell_2}^2\right] = \|\boldsymbol{B}\|_F^2 + \|\boldsymbol{w}\|_{\ell_2}^2 \frac{d+1}{2} - \sqrt{\frac{2}{\pi}}\left(\|\boldsymbol{w}\|_{\ell_2} \operatorname{trace}(\boldsymbol{B}) + \frac{\boldsymbol{w}^\mathsf{T}\boldsymbol{B}\boldsymbol{w}}{\|\boldsymbol{w}\|_{\ell_2}}\right). \tag{B.8}$$

With this characterization, we next proceed to calculate the optimal objective value of (B.4). We start by minimizing over $\boldsymbol{w}$. To do this, we first fix $\|\boldsymbol{w}\|_{\ell_2} = \alpha$, and optimize over the direction of $\boldsymbol{w}$, and then optimize over $\alpha$. Note that $\boldsymbol{w}^\mathsf{T}\boldsymbol{B}\boldsymbol{w} = \boldsymbol{w}^\mathsf{T}(\boldsymbol{B} + \boldsymbol{B}^\mathsf{T})/2\boldsymbol{w}$. Since $(\boldsymbol{B} + \boldsymbol{B}^\mathsf{T})/2$ is symmetric, the maximum is achieved when $\boldsymbol{w}$ s in the direction of its maximum eigenvalue. Define $\lambda_{\max}^B$ as the maximum eigenvalue of $(\boldsymbol{B} + \boldsymbol{B}^\mathsf{T})/2$. We then have

$$\min_{\boldsymbol{w}} \mathbb{E}\left[\|\boldsymbol{B}\boldsymbol{x} - \sigma(\boldsymbol{w}^\mathsf{T}\boldsymbol{x})\boldsymbol{x}\|_{\ell_2}^2\right] = \min_{\alpha \geq 0} \|\boldsymbol{B}\|_F^2 + \alpha^2 \frac{d+1}{2} - \sqrt{\frac{2}{\pi}}\alpha\left(\operatorname{trace}(\boldsymbol{B}) + \lambda_{\max}^B\right)$$

$$= \|\boldsymbol{B}\|_F^2 - \frac{(\operatorname{trace}(\boldsymbol{B}) + \lambda_{\max}^B)^2}{\pi(d+1)}. \tag{B.9}$$

We next continue with minimization over $\boldsymbol{M}, \boldsymbol{D}$. Since we want to derive upper bound on the minimum objective value, we consider two choices of $(\boldsymbol{M}, \boldsymbol{D})$ motivated by the analysis of GRN-v1 and GRN-v2.

- Choice 1: Similar to the analysis of GRN-v1, we set $\boldsymbol{M} = \boldsymbol{U}_{r*}[\boldsymbol{\Sigma}_{r*}\boldsymbol{V}_{r*}^\mathsf{T} + (1 - \alpha)\boldsymbol{U}_{r*}^\mathsf{T}]$ and $\boldsymbol{D} = \alpha\boldsymbol{I}$ with $\alpha = 1 + \frac{1}{d-r_*}\operatorname{trace}(\boldsymbol{\Delta})$. With these choices we have

$$\boldsymbol{B} = \boldsymbol{A} - \boldsymbol{M} - \boldsymbol{D}$$
$$= \boldsymbol{A} - \boldsymbol{I} - \boldsymbol{M} + (1 - \alpha)\boldsymbol{I}$$
$$= \boldsymbol{\Delta} - (1 - \alpha)\boldsymbol{U}_{r*}\boldsymbol{U}_{r*}^\mathsf{T} + (1 - \alpha)\boldsymbol{I}$$
$$= \boldsymbol{\Delta} + (1 - \alpha)\boldsymbol{U}_{r*,\perp}\boldsymbol{U}_{r*,\perp}^\mathsf{T}$$
$$= \boldsymbol{\Delta} - \frac{1}{d - r_*}\operatorname{trace}(\boldsymbol{\Delta})\boldsymbol{U}_{r*,\perp}\boldsymbol{U}_{r*,\perp}^\mathsf{T}.$$

In addition, $\left\|\boldsymbol{U}_{r*,\perp}\boldsymbol{U}_{r*,\perp}^\mathsf{T}\right\|_F^2 = d - r_*$ and $\operatorname{trace}(\boldsymbol{\Delta}^\mathsf{T}\boldsymbol{U}_{r*,\perp}\boldsymbol{U}_{r*,\perp}^\mathsf{T}) = \operatorname{trace}(\boldsymbol{\Delta})$. Hence,

$$\|\boldsymbol{B}\|_F^2 = \|\boldsymbol{\Delta}\|_F^2 - \frac{1}{d - r_*}\operatorname{trace}(\boldsymbol{\Delta})^2.$$

Furthermore, $\operatorname{trace}(\boldsymbol{B}) = 0$. Using these identities in (B.9) we obtain that the optimum objective value of (B.4) satisfies the following:

$$OPT \leq \|\boldsymbol{\Delta}\|_F^2 - \frac{1}{d - r_*}\operatorname{trace}(\boldsymbol{\Delta})^2 - \frac{\nu_{\max}^2}{\pi(d+1)}, \tag{B.10}$$

with $\nu_{\max}$ denoting the maximum eigenvalue of $\frac{\boldsymbol{\Delta} + \boldsymbol{\Delta}^\mathsf{T}}{2} - \frac{1}{d - r_*}\operatorname{trace}(\boldsymbol{\Delta})\boldsymbol{U}_{r*,\perp}\boldsymbol{U}_{r*,\perp}^\mathsf{T}$.

- Choice 2: Similar to the analysis of GRN-v2, we set $\boldsymbol{U}_{r*}\boldsymbol{\Sigma}_{r*}\boldsymbol{V}_{r*}^\mathsf{T}$ and $\boldsymbol{D} = \operatorname{diag}(\boldsymbol{I} + \boldsymbol{\Delta})$. This way we have

$$\boldsymbol{B} = \boldsymbol{A} - \boldsymbol{M} - \boldsymbol{D}$$
$$= \boldsymbol{A} - \boldsymbol{I} - \boldsymbol{M} + \boldsymbol{I} - \boldsymbol{D}$$
$$= \boldsymbol{\Delta} + \boldsymbol{I} - \boldsymbol{D}$$
$$= \boldsymbol{\Delta} - \operatorname{diag}(\boldsymbol{\Delta}).$$

Hence, $\|\boldsymbol{B}\|_F^2 = \|\boldsymbol{\Delta}\|_F^2 - \sum_{i=1}^d \Delta_{ii}^2$ and $\text{trace}(\boldsymbol{B}) = 0$. Using these identities in (B.11) we obtain that the optimum objective value of (B.4) satisfies the following:

$$OPT \leq \|\boldsymbol{\Delta}\|_F^2 - \sum_{i=1}^d \Delta_{ii}^2 - \frac{\tilde{\nu}_{\max}^2}{\pi(d+1)},$$

with $\tilde{\nu}$ denoting the maximum eigenvalue of $\frac{\boldsymbol{\Delta}+\boldsymbol{\Delta}^{\mathsf{T}}}{2} - \text{diag}(\boldsymbol{\Delta})$.

Combining the bound from the two cases, we get

$$\text{Err}^*(\mathcal{C}_{\text{GRN}-\text{v3}}) \leq OPT \leq \|\boldsymbol{\Delta}\|_F^2 - \max\left\{\frac{1}{d-r_*}\text{trace}(\boldsymbol{\Delta})^2 + \frac{\nu_{\max}^2}{\pi(d+1)}, \sum_{i=1}^d \Delta_{ii}^2 + \frac{\tilde{\nu}_{\max}^2}{\pi(d+1)}\right\},$$

which completes the proof of theorem.

*Proof.* (Lemma B.1) Since the distribution of $\boldsymbol{x}$ is rotation invariant, without loss of generality we assume $\boldsymbol{w} = \|\boldsymbol{w}\|_{\ell_2} \boldsymbol{e}_1$. We then have

$$\mathbb{E}[\sigma(\boldsymbol{w}^{\mathsf{T}}\boldsymbol{x})^2 \|\boldsymbol{x}\|_{\ell_2}^2] = \|\boldsymbol{w}\|_{\ell_2}^2 \mathbb{E}[\sigma(x_1)^2(x_1^2 + \|\boldsymbol{x}_{\sim 1}\|_{\ell_2}^2)] \tag{B.11}$$

where $\boldsymbol{x}_{\sim 1} = (x_2, \ldots, x_d)$. We have

$$\mathbb{E}[\sigma(x_1)^2 x_1^2] = \mathbb{E}[x_1^4 \mathbf{1}(x_1 \geq 0)] = \frac{1}{2}\mathbb{E}[x_1^4] = \frac{3}{2}.$$

Also, since $\boldsymbol{x}_{\sim 1}$ is independent of $x_1$ we have

$$\mathbb{E}[\sigma(x_1)^2 \|\boldsymbol{x}_{\sim 1}\|_{\ell_2}^2] = \mathbb{E}[\sigma(x_1)^2]\mathbb{E}[\|\boldsymbol{x}_{\sim 1}\|_{\ell_2}^2] = \frac{d-1}{2}.$$

Combining the last three equations, we get

$$\mathbb{E}[\sigma(\boldsymbol{w}^{\mathsf{T}}\boldsymbol{x})^2 \|\boldsymbol{x}\|_{\ell_2}^2] = \|\boldsymbol{w}\|_{\ell_2}^2 \frac{d+1}{2}.$$

To show the other claim, we note that

$$\mathbb{E}[\sigma(x_1)x_i x_j] = \begin{cases} 0 & \text{if } i \neq j \\ = \frac{1}{2}\mathbb{E}[|x_1|]\mathbb{E}[x_i^2] = \frac{1}{\sqrt{2\pi}}, & \text{if } i = j \neq 1, \\ = \frac{1}{2}\mathbb{E}[|x_1|^3] = \sqrt{\frac{2}{\pi}} & \text{if } i = j = 1. \end{cases} \tag{B.12}$$

Therefore in matrix form we have $\mathbb{E}[\sigma(x_1)\boldsymbol{x}\boldsymbol{x}^{\mathsf{T}}] = \frac{1}{\sqrt{2\pi}}(\boldsymbol{I} + \boldsymbol{e}_1\boldsymbol{e}_1^{\mathsf{T}})$. This completes the proof of (B.7) as by rotation invariance of distribution of $\boldsymbol{x}$ we can assume $\boldsymbol{w} = \|\boldsymbol{w}\|_{\ell_2} \boldsymbol{e}_1$. $\square$

## C. Proof of Theorem 4.3

Let $p := 2dr_*$ where we recall that $r_* = \sum_{\ell=1}^T r_\ell$. Note that $p$ is the number of parameters for ResNet with $T$ layers and ranks $r_t$ for each layer $t$. We will compare the test error of GRN-v1 and ResNet with $p$ number of parameters. This corresponds to a model in GRN-v1 with $T'$ layers such that $2d\sum_{\ell=1}^{T'} r_\ell + T'(T'-1)/2 = p$. We set the shorthand $r'_* := \sum_{\ell=1}^{T'} r_\ell$ and let $\sigma_1 \geq \ldots \geq \sigma_d$ be the singular values of $\boldsymbol{A} - \boldsymbol{I}$. By Theorem 4.2 we have

$$\text{ER}^*(\mathcal{C}_{\text{res}}) = \sum_{i=r_*+1}^d \sigma_i^2, \quad \text{ER}^*(\mathcal{C}_{\text{GRN}-\text{v1}}) \leq \sum_{i=r'_*+1}^d \sigma_i^2 - \frac{1}{d-r'_*}\left(\sum_{i=r'_*+1}^d \sigma_i\right)^2$$

Therefore, $\mathsf{ER}^*(\mathcal{C}_{\mathrm{GRN-v1}}) < \mathsf{ER}^*(\mathcal{C}_{\mathrm{res}})$ if the following holds:

$$\sum_{i=r'_*+1}^{r_*} \sigma_i^2 \leq \frac{1}{d-r'_*}\Big(\sum_{i=r'_*+1}^{d} \sigma_i\Big)^2 \tag{C.1}$$

(Note that $r'_* < r_*$ since $T' < T$). However note that the left hand side of this condition is upper bounded by

$$\sum_{i=r'_*+1}^{r_*} \sigma_i^2 \leq (r_* - r'_*)\lambda_{\max}^2$$

Additionally, the right-hand side of the condition is lower bounded by

$$\frac{1}{d-r'_*}\Big(\sum_{i=r'_*+1}^{d} \sigma_i\Big)^2 \geq (d-r'_*)\lambda_{\min}^2\,.$$

So a sufficient condition for (C.1) is that

$$(r_* - r'_*)\lambda_{\max}^2 \leq (d-r'_*)\lambda_{\min}^2\,.$$

Writing it in terms of $\kappa$, we need

$$r_* \leq d\kappa^2 + r'_*(1-\kappa^2). \tag{C.2}$$

Our next lemma gives alower bound on $r'_*$.

**Lemma C.1.** *Consider a standard Resnet model with collective rank $r_*$, and also a GRN-v1 model with collective rank $r'$, a GRN-v2 model with collective rank $r''$, and a GRN-v3 model with collective rank $r'''$, which have the same number of parameters as in the standard Resent model. We then have*

$$r_* - (\sqrt{d+r_*} - \sqrt{d})^2 \leq r'_* \leq r_*\,, \tag{C.3}$$
$$r_* - (\sqrt{1+r_*} - 1)^2 \leq r''_* \leq r_*\,, \tag{C.4}$$
$$r_* - (\sqrt{1.6+r_*} - 1)^2 \leq r'''_* \leq r_*\,. \tag{C.5}$$

Using Lemma C.1, condition C.2 is satisfied provided that

$$r_* \leq d\kappa^2 + (1-\kappa^2)\Big[r_* - (\sqrt{d+r_*} - \sqrt{d})^2\Big]\,.$$

Solving the above inequality for $r_*/d$ and after some algebraic calculation, we simplify the above inequality as follows:

$$\frac{r_*}{d} \leq (1 + \kappa(\sqrt{\kappa^2+1} - \kappa))^2 - 1\,.$$

For GRN-v2, the argument goes along the same lines. Fixing number of parameters to $p$, this corresponds to a model in GRN-v2 with $T''$ layers such that $2d\sum_{\ell=1}^{T''} r_\ell + dT''(T''-1)/2 = p$. We use the shorthand $r''_* := \sum_{\ell=1}^{T''} r_\ell$. By Theorem 4.2,

$$\mathsf{ER}^*(\mathcal{C}_{\mathrm{GRN-v2}}) = \|\boldsymbol{\Delta}\|_F^2 - \frac{1}{d-r''_*}\mathrm{trace}(\boldsymbol{\Delta})^2$$

$$= \sum_{i=r''_*+1}^{d} \sigma_i^2 - \frac{1}{d-r''_*}\Big(\sum_{i=r''_*+1}^{d} \sigma_i\Big)^2\,.$$

Following the same argument as the one for GRN-v1 (replacing $r'_*$ with $r''_*$) we derive that GRN-v2 achieves a better trade-off than standard ResNet, if

$$r_* \leq d\kappa^2 + r''_*(1-\kappa^2). \tag{C.6}$$

(Note that this is analogous to (C.2) where $r'_*$ is replaced by $r''_*$.)

Using Lemma C.1, condition C.6 is satisfied provided that

$$r_* \leq d\kappa^2 + (1 - \kappa^2)\Big[r_* - (\sqrt{1 + r_*} - 1)^2\Big].$$

By some algebraic calculation, this inequality can be simplified to

$$r_* \leq (1 + \kappa(\sqrt{\kappa^2 + d} - \kappa))^2 - 1.$$

We next proceed with the case of GRN-v3. By Theorem 4.2 we have

$$\mathsf{ER}^*(\mathcal{C}_{\mathrm{GRN-v3}}) \leq \|\mathbf{\Delta}\|_F^2 - \frac{1}{d - r_*}\mathrm{trace}(\mathbf{\Delta})^2 - \frac{\nu_{\max}^2}{\pi(d + 1)}$$

with $\nu_{\max}$ denoting the maximum eigenvalue of $\frac{\mathbf{\Delta} + \mathbf{\Delta}^\mathsf{T}}{2} - \frac{1}{d - r_*}\mathrm{trace}(\mathbf{\Delta})\mathbf{U}_{r''_*,\perp}\mathbf{U}_{r''_*,\perp}^\mathsf{T}$. Rewriting this bound in terms of eigenvalues we get

$$\mathsf{ER}^*(\mathcal{C}_{\mathrm{GRN-v3}}) \leq \sum_{i=r'''_*+1}^{d} \sigma_i^2 - \frac{1}{d - r'''_*}\left(\sum_{i=r'''_*+1}^{d} \sigma_i\right)^2 - \frac{1}{\pi(d + 1)}\left(\sigma_{r'''_*+1} - \frac{1}{d - r'''_*}\sum_{i=r'''_*+1}^{d} \sigma_i\right)^2.$$

Here we used the fact that the $\mathbf{U}_{r'''_*,\perp}$ is the eigenspace of $\mathbf{\Delta}$ and its eigenvalues are $\sigma_{r'''_*+1} \geq \ldots \geq \sigma_d$. To lighten the notation, we use the shorthand $\sigma := \sigma_{r'''_*+1}$ and $A := \sum_{i=r'''_*+1}^{d} \sigma_i$. Then in order to have $\mathsf{ER}^*(\mathcal{C}_{\mathrm{GRN-v3}}) < \mathsf{ER}^*(\mathcal{C}_{\mathrm{res}})$, it suffices to have

$$\sum_{i=r'''_*+1}^{r_*} \sigma_i^2 \leq \frac{1}{d - r'''_*}A^2 - \frac{1}{\pi(d + 1)}(\sigma - \frac{1}{d - r'''_*}A)^2.$$

Note that the left-hand side is upper bounded by $(r_* - r'''_*)\sigma$. In addition, this is quadratic inequality in $A$. Solving this inequality for $A$ this corresponds to the following:

$$\frac{A}{\sigma} \geq \frac{\frac{1}{\pi(d+1)(d-r'''_*)} + \sqrt{\frac{r_*-r'''_*}{d-r'''_*} + \frac{r_*-r'''_*}{\pi(d+1)(d-r'''_*)^2} - \frac{1}{\pi(d+1)(d-r'''_*)}}}{\frac{1}{d-r'''_*} + \frac{1}{\pi(d+1)(d-r'''_*)^2}}.$$

Define the shorthand $\xi := \frac{1}{\pi(d+1)(d-r'''_*)}$. Then the above can be written as

$$\frac{A}{\sigma} \geq \frac{\xi + \sqrt{\frac{r_*-r'''_*}{d-r'''_*}(1 + \xi) - \xi}}{\frac{1}{d-r_*}(1 + \xi)}. \tag{C.7}$$

We also have $A \geq (d - r'''_*)\lambda_{\min}$ and $\sigma \leq \lambda_{\max}$, so $A/\sigma \geq (d - r'''_*)\kappa$. Hence, the above condition holds if

$$\kappa \geq \frac{\xi + \sqrt{\frac{r_*-r'''_*}{d-r'''_*}(1 + \xi) - \xi}}{1 + \xi}. \tag{C.8}$$

It is easy to verify that the right-hand side is decreasing in $\xi$. In addition, by definition of $\xi$ and since $r'''_* < r_* \leq d$, we have $\xi \geq \xi_0 := \frac{1}{\pi(d^2-1)}$. So a sufficient condition for (C.8) is

$$\kappa \geq \frac{\xi_0 + \sqrt{\frac{r_*-r'''_*}{d-r'''_*}(1 + \xi_0) - \xi_0}}{1 + \xi_0} \tag{C.9}$$

or equivalently,

$$\frac{(\kappa(1 + \xi_0) - \xi_0)^2 + \xi_0}{1 + \xi_0} \geq \frac{r_* - r'''_*}{d - r'''_*}.$$

Define $\eta := \sqrt{\frac{(\kappa(1+\xi_0)-\xi_0)^2+\xi_0}{1+\xi_0}}$. Rewriting the above inequality we need

$$r_*'''(1-\eta^2) + d\eta^2 \geq r_* \,. \tag{C.10}$$

Next, by Lemma C.1, we have $r_* - r_*''' \leq (\sqrt{1.6 + r_*} - 1)^2$, by which a sufficient condition for (C.10) is as follows:

$$\left[ r_* - (\sqrt{1.6 + r_*} - 1)^2 \right] (1-\eta^2) + d\eta^2 \geq r_* \,.$$

By some algebraic calculation, this inequality can be simplified to

$$r_* \leq (1 + \eta(\sqrt{\eta^2 + d} - \eta))^2 - 1.6$$

This completes the proof of the first item in the theorem statement.

To prove the second item in the theorem statement, we write

$$
\begin{aligned}
G_1 := \mathsf{ER}^*(\mathcal{C}_{\mathrm{res}}) - \mathsf{ER}^*(\mathcal{C}_{\mathrm{GRN-v1}}) &\geq \frac{1}{d - r_*'} \Big( \sum_{i=r_*'+1}^{d} \sigma_i \Big)^2 - \sum_{i=r_*'+1}^{r_*} \sigma_i^2 \\
&\geq (d - r_*')\lambda_{\min}^2 - (r_* - r_*')\lambda_{\max}^2 \\
&= d\lambda_{\min}^2 - r_*\lambda_{\max}^2 + r_*'(\lambda_{\max}^2 - \lambda_{\min}^2) \\
&\geq d\lambda_{\min}^2 - r_*\lambda_{\max}^2 + (r_* - (\sqrt{d + r_*} - \sqrt{d})^2)(\lambda_{\max}^2 - \lambda_{\min}^2) \\
&= (d - r_*)\lambda_{\min}^2 - (\sqrt{d + r_*} - \sqrt{d})^2(\lambda_{\max}^2 - \lambda_{\min}^2) \,,
\end{aligned}
$$

where in the last inequality we used Lemma C.1.

A similar bound can be derived for GRN-v2, replacing $r_*'$ with $r_*''$ in the argument. Specifically, we have

$$
\begin{aligned}
G_2 := \mathsf{ER}^*(\mathcal{C}_{\mathrm{res}}) - \mathsf{ER}^*(\mathcal{C}_{\mathrm{GRN-v2}}) &\geq d\lambda_{\min}^2 - r_*\lambda_{\max}^2 + r_*''(\lambda_{\max}^2 - \lambda_{\min}^2) \\
&\geq d\lambda_{\min}^2 - r_*\lambda_{\max}^2 + (r_* - (\sqrt{1 + r_*} - 1)^2)(\lambda_{\max}^2 - \lambda_{\min}^2) \\
&= (d - r_*)\lambda_{\min}^2 - (\sqrt{1 + r_*} - 1)^2)(\lambda_{\max}^2 - \lambda_{\min}^2) \,,
\end{aligned}
$$

where in the last inequality we used the lower bound given for $r''$ in Lemma C.1.

For GRN-v3, we have

$$
\begin{aligned}
G_3 := \mathsf{ER}^*(\mathcal{C}_{\mathrm{res}}) - \mathsf{ER}^*(\mathcal{C}_{\mathrm{GRN-v3}}) &\geq \frac{1}{d - r_*'''} \Big( \sum_{i=r_*'''+1}^{d} \sigma_i \Big)^2 + \frac{\lambda_{\max}^2}{\pi(d+1)} - \sum_{i=r_*'''+1}^{r_*} \sigma_i^2 \\
&= \frac{1}{d - r_*'''} A^2 + \frac{1}{\pi(d+1)}(\sigma - \frac{1}{d - r_*'''} A)^2 - \sum_{i=r_*'''+1}^{r_*} \sigma_i^2 \\
&\geq \frac{1}{d - r_*'''} A^2 + \frac{1}{\pi(d+1)} \Big( \frac{\sigma^2}{2} - \frac{1}{(d - r_*''')^2} A^2 \Big) - (r_* - r_*''')\sigma^2 \\
&\geq \frac{1}{d - r_*'''} \Big( 1 - \frac{1}{\pi(d+1)(d - r_*''')} \Big) A^2 - \Big( r_* - r_*''' - \frac{1}{2\pi(d+1)} \Big)\sigma^2 \,,
\end{aligned}
$$

where we recall the shorthand $A := \sum_{i=r_*'''+1}^{d} \sigma_i$ and $\sigma := \sigma_{r_*'''+1}$. In the second inequality above, we used the fact that $\sigma_{r_*'''+1}, \ldots, \sigma_{r_*} \leq \sigma_{r_*'''+1} = \sigma$, along with the inequality $(a - b)^2 \geq a^2/2 - b^2$.

Noting that $A \geq (d - r_*''')\lambda_{\min}$ and $\sigma \leq \lambda_{\max}$, we continue from the above chain of inequalities, as follows:

$$
\begin{aligned}
G_3 &= \left(d - r''' - \frac{1}{\pi(d+1)}\right)\lambda_{\min}^2 - \left(r_* - r''' - \frac{1}{2\pi(d+1)}\right)\lambda_{\max}^2 \\
&\geq \left(d - \frac{1}{\pi(d+1)}\right)\lambda_{\min}^2 - \left(r_* - \frac{1}{2\pi(d+1)}\right)\lambda_{\max}^2 + r'''(\lambda_{\max}^2 - \lambda_{\min}^2) \\
&\stackrel{(a)}{\geq} \left(d - \frac{1}{\pi(d+1)}\right)\lambda_{\min}^2 - \left(r_* - \frac{1}{2\pi(d+1)}\right)\lambda_{\max}^2 + (r_* - (\sqrt{1.6 + r_*} - 1)^2)(\lambda_{\max}^2 - \lambda_{\min}^2) \\
&= \left(d - \frac{1}{\pi(d+1)} - r_*\right)\lambda_{\min}^2 + \frac{1}{2\pi(d+1)}\lambda_{\max}^2 - (\sqrt{1.6 + r_*} - 1)^2(\lambda_{\max}^2 - \lambda_{\min}^2),
\end{aligned}
$$

where we used Lemma C.1 in step $(a)$.

### C.1. Proof of Lemma C.1

A standard Resnet model with collective rank $r_*$ has $2dr_*$ number of parameters. A model in GRN-v1 with collective rank $r_*'$ has $2dr_*' + T'(T'-1)/2$ parameters. Therefore, by assumption

$$2dr_*' + T'(T'-1)/2 = 2dr_*. \tag{C.11}$$

Since each layer has rank at least one, we also have $r_*' \geq T'$. We define the shorthand $\xi = \sqrt{T'(T'-1)}$ (so $r' \geq \xi$). Combining these two inequalities and writing them in terms of $\xi$, we get

$$2d\xi + \xi^2/2 \leq 2dr_*.$$

Solving this inequality for $\xi$ we get $\xi \leq 2\sqrt{d^2 + dr_*} - 2d$. Using this bound in (C.11) we get

$$2dr_* \leq 2dr_*' + 2(\sqrt{d^2 + dr_*} - d)^2.$$

Simplifying this inequality, we arrive at

$$r_* - (\sqrt{d + r_*} - \sqrt{d})^2 \leq r_*'.$$

The upper bound $r_*' \leq r_*$ also follows simply from (C.11).

For GRN-v2 model we follow the same argument. A model in GRN-v2 with collective rank $r''$ has $2dr_*'' + dT''(T''-1)/2$ parameters and so

$$2r_*'' + T''(T''-1)/2 = 2r_*. \tag{C.12}$$

Since each layer has rank at least one, we also have $r_*'' \geq T''$. Define the shorthand $\xi' := \sqrt{T''(T''-1)}$. Combining the previous two equation, we get

$$2\xi' + \xi'^2/2 \leq 2r_*.$$

Solving this inequality for $\xi'$ we get $\xi' \leq 2\sqrt{1 + r_*} - 2$. Using this bound back in (C.12) we obtain

$$2r_* \leq 2r_*'' + 2(\sqrt{1 + r_*} - 1)^2.$$

This simplifies to $r_* - (\sqrt{1 + r_*} - 1)^2 \leq r_*''$.

We follow the same argument for GRN-v3. A model in GRN-v3 with collective rank $r'''$ has $2dr_*''' + dT'''(T'''-1)/2 + dT'''$ parameters and so

$$2r_*''' + T'''(T''' + 1)/2 = 2r_*. \tag{C.13}$$

Since each layer has rank at least one, we also have $r_*''' \geq T'''$. Hence,

$$T'''^2 + 5T''' = 4r_* \leq 0.$$

Solving for $T'''$ we get $T''' \leq 1/2(\sqrt{25 + 16r_*} - 5)$. Using this bound in (C.13), we have

$$
\begin{aligned}
r_*''' &\geq r_* - \frac{1}{16}(\sqrt{25 + 16r_*} - 5)(\sqrt{25 + 16r_*} - 3) \\
&= r_* - \frac{1}{16}(40 + 16r_* - 8\sqrt{25 + 16r_*}) \\
&\geq r_* - \frac{1}{16}\left(\sqrt{25 + 16r_*} - 4\right)^2 \\
&\geq r_* - \left(\sqrt{1.6 + r_*} - 1\right)^2 .
\end{aligned}
$$

The upper bound $r_*''' < r_*$ follows easily from (C.13).

## D. Proof of Proposition 4.4

The result follows from conditions (C.2), (C.6) and (C.10) which respectively provide sufficient conditions for GRN-v1, GRN-v2 and GRN-v3 to achieve smaller test error than a ResNet model, with the same number of parameters.

## E. Proof of Proposition 4.5

The proof is similar to the linear case by induction on $T$. Note that for showing this direction ($\mathcal{C}_{\text{base}}, \mathcal{C}_{\text{res}}, \mathcal{C}_{\text{GRN}-v1}, \mathcal{C}_{\text{GRN}-v2}$ being a subset of the rank constrained functions) we only used the following two properties of the rank function which holds also for the Bottleneck rank: $\text{rank}(f \circ g) \leq \min\{\text{rank}(f), \text{rank}(g)\}$ and $\text{rank}(f + g) \leq \text{rank}(f) + \text{rank}(g)$.

## F. Training time versus perplexity on the LM1B dataset

This appendix provides additional results on training time versus perplexity for DCA models. Figure 8 shows the training time-perplexity trade-off for 12, 24, and 36 layer transformer and DCA models trained on the LM1B dataset. The figure shows that DCA achieves a better perplexity for a given training time (except for the first few training steps of the 36-layer model). Thus, highlighting the training efficiency of DCA.

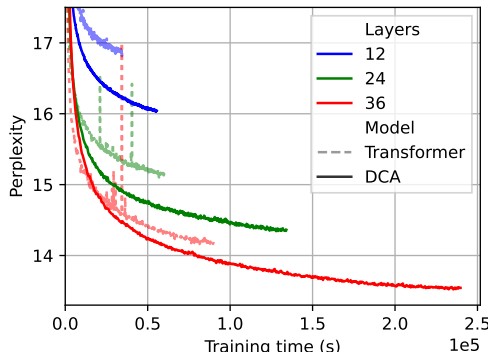

*Figure 8.* Perplexity on LM1B pre-training versus the training time with transformer and DCA models of various depths.

## G. Steps versus perplexity on the C4 dataset

This appendix provides additional results on training steps versus perplexity for 8-DCA models. Figure 9 shows the training steps-perplexity trade-off for 75M, 179M, and 449M parameter transformer and 8-DCA models trained on the C4 dataset. The results show the improved model convergence and quality of DCA.

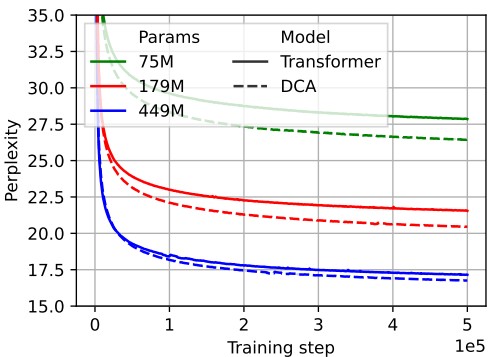

*Figure 9.* Perplexity on C4 pre-training versus the number of steps with transformer and 8-DCA models of various sizes.

*Table 9.* Perplexity (PPL) on LM1B with and without the model inputs for 6-layer GRN-v3.

| METHOD | PPL |
|---|---|
| TRANSFORMER | 20.878 |
| GRN-V3 (LAST 4 LAYERS) | 20.301 |
| GRN-V3 (MODEL INPUTS + LAST 3 LAYERS) | **20.227** |

# H. Distribution of learned weights

Figure 10 shows the distribution of the learned bias values for each GRN-v3 instance of a 30-layer model. The layers tend to weight the inputs and the last few layers the most and frequently assign a negative bias for the intermediate layers, indicating that the layers are filtered out as a result of the ReLU activation. In Table 9, we show that indeed the GRN-v3 model perplexity improves as the model inputs are included in addition to the last few layer outputs.

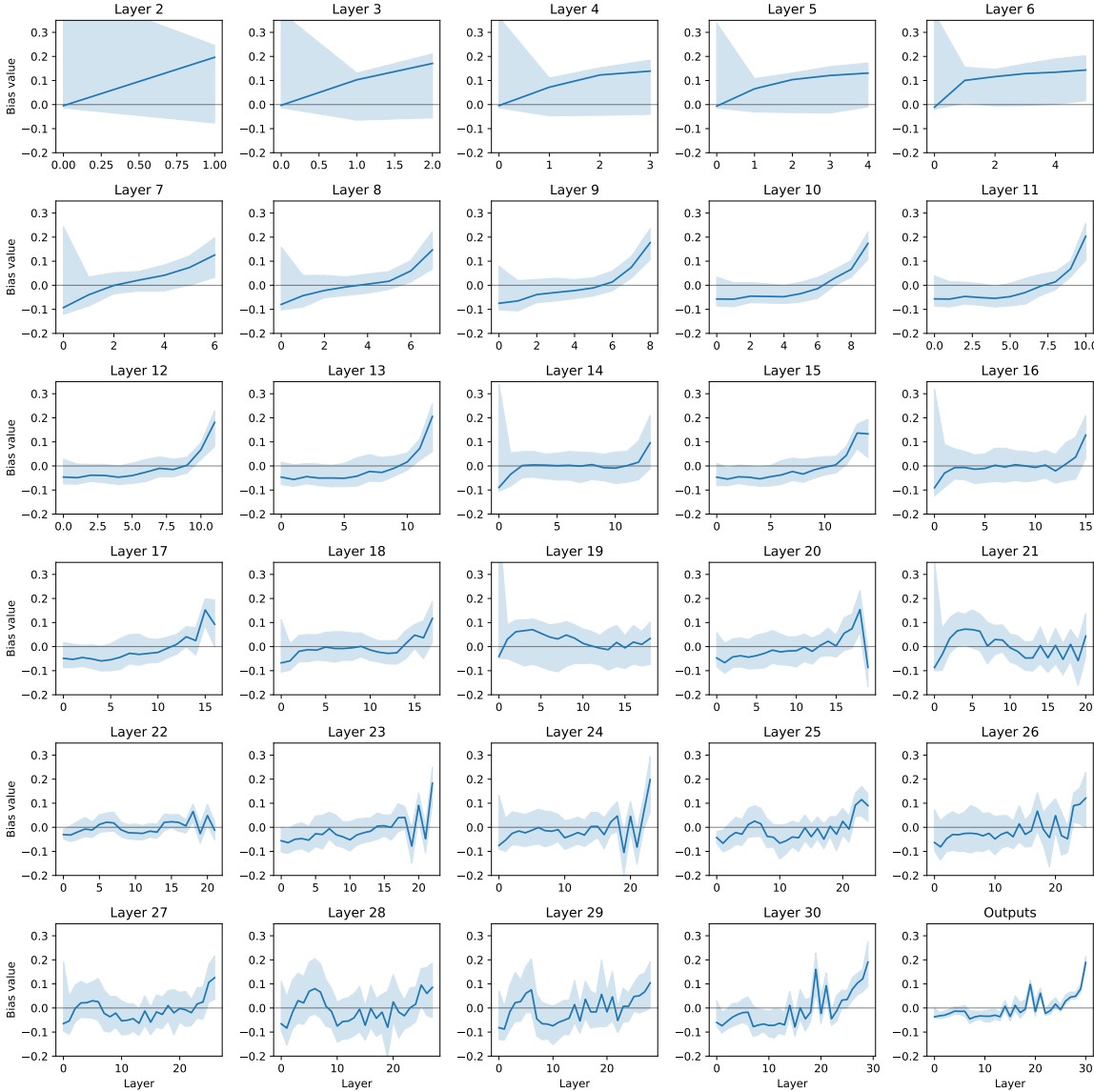

*Figure 10.* Distribution of learned bias values on LM1B pre-training with a 30 layer GRN-v3 transformer model. The solid line indicates the median value and the shaded area represents the 90th percentile.

