# OpenReview forum: "DeepCrossAttention: Supercharging Transformer Residual Connections"
_ICML.cc/2025/Conference — ICML 2025 poster_

### Official Review · Reviewer_rA8T · 2025-03-08

**Overall Recommendation:** 3

**Summary:**

The authors propose DeepCrossAttention (DCA), a novel method that enhances residual connections in transformer architectures. In standard transformers, if we denote the input and output of the i'th block as $x_i$ and $y_i$ respectively, vanilla residual connections simply use $x_i = \sum_{j < i} y_j$. DCA instead employs learnable, input-dependent weights to dynamically combine layer outputs: $x_i = \sum_{j < i} y_j \cdot (\text{ReLU}(\langle y_j, w_{ij} \rangle) + b_{ij})$ where $w_{ij}$ and $b_{ij}$ are parameter vectors.

For attention layers specifically, DCA creates queries, keys, and values by independently combining previous layer outputs using separate parameter vectors, allowing for richer interactions between layers at different depths.

## Theoretical Contributions
The authors analyze stacked low-rank linear projections, demonstrating that DCA achieves a better accuracy-model size trade-off when the ratio of collective layer ranks to ambient dimension falls below a critical threshold. They extend this analysis to nonlinear models using the concept of bottleneck rank.

## Empirical Contributions
Extensive language modeling experiments on LM1B and C4 datasets demonstrate that DCA:
- Achieves better perplexity for a given parameter budget
- Reaches equivalent model quality up to 3x faster
- Exhibits improved training stability with fewer loss spikes
- Adds only a negligible number of parameters (approximately 0.2%)

## Strengths
1. The method is elegant in its simplicity while delivering substantial convergence improvements, particularly for models with smaller hidden dimensions
2. Comprehensive empirical validation across multiple scales (75M to 449M parameters) using various metrics
3. Thorough ablation studies isolate the contribution of each proposed component
4. Clear improvements over related methods (LAuReL, DenseFormer, Hyper-Connections)

## Weaknesses
1. Experimental validation is limited to language modeling tasks; testing on other modalities (vision, audio) would strengthen the paper's claims about the method's generalizability
2. The theoretical analysis, while sound, is less compelling than the experimental results; the authors could have expanded the empirical evaluations section instead

**Claims And Evidence:**

Yes, the claims are supoorted theoretically and empirically.

**Essential References Not Discussed:**

-

**Experimental Designs Or Analyses:**

Yes, the experiments are sound.

**Methods And Evaluation Criteria:**

Yes LM on C4 and LM1B makes sense but the evaluation could be further expanded.

**Other Comments Or Suggestions:**

-

**Other Strengths And Weaknesses:**

-

**Questions For Authors:**

See weaknesses

**Relation To Broader Scientific Literature:**

The proposed method builds on related work and the novelity over past work is not significant - however the authors do show improved reuslts over the methods that they build on.

**Theoretical Claims:**

I checked the correctness of proofs and did not find any issues.

---

> ### Author Rebuttal · Authors · 2025-03-31
>
> We thank the reviewer for taking the time to review our manuscript. We address the two concerns raised by the reviewer below.
>
> > Experimental validation is limited to language modeling tasks; testing on other modalities (vision, audio) would strengthen the paper's claims about the method's generalizability.
>
> Based on the reviewer’s suggestion we have performed additional experiments on ImageNet classification using vision transformers. Since the ViT model is also transformer-based, we were able to incorporate DCA the same way as for the language models presented in the manuscript. We present our results on the ViT-S/16 model from https://github.com/google-research/big_vision (22M params):
>
> | Method    | Training loss | Accuracy |
> | --- | --- | --- |
> | ViT  | 0.5698   | 76.4 |
> | ViT+DCA |  **0.5284** | **77.1** |
>
> This indicates that our results generalize to the vision domain.
>
> > The theoretical analysis, while sound, is less compelling than the experimental results; the authors could have expanded the empirical evaluations section instead.
>
> In response to the reviewer’s feedback, we have significantly expanded the empirical results section of our manuscript. The updated version will include the ImageNet results presented above, as well as additional comparisons with prior work using larger-scale models (see also the response to reviewer nMe5) to provide a more comprehensive empirical evaluation of our method.
>
> We are confident that these additional experiments and clarifications address the reviewer's concerns and further strengthen the paper.

---

> > ### Comment · Reviewer_rA8T · 2025-04-02
> >
> > Thanks - I am maintaining my score.

---

### Official Review · Reviewer_zWQ6 · 2025-03-10

**Overall Recommendation:** 3

**Summary:**

The authors introduce learnable residual connections to improve over standard residual connections used in ResNets and transformers. They highlight that simple residual connections struggle to recover the input (learn the identity function) on toy examples and their proposed learnable residual connections can overcome this problem. Theoretical analysis on a low rank linear model shows that their proposed method obtains lower risk given that the rank of the task is small enough. Further experiments on transformers are provided to highlight to empirically evaluate the method.

######## update after rebuttal #########

The authors have clarified my questions, i will keep my score.

**Claims And Evidence:**

Yes, the proposed model architecture has been validated empirically.

**Essential References Not Discussed:**

Most of the relevant literature is discussed.

**Experimental Designs Or Analyses:**

The experiments are designed to verify the performance of the newly proposed GRN architecture.

**Methods And Evaluation Criteria:**

The method is evaluated across multiple datasets.

**Other Comments Or Suggestions:**

See above

**Other Strengths And Weaknesses:**

Strengths

1. The proposed method is simple and can provide significant improvements as well as faster training as shown in the experiments.
2. The authors also provide some theoretical insights to validate their method, and show that as long as the rank of the target task is small enough, the GRN model can achieve lower risk.


Weaknesses

1. Most efficiency gains seem to occur by using the first and last-k layer outputs in the GRN, for k=2. Moreover the perplexity gains from increasing k further are limited. This implies that not all layer representations contribute to the residual connection in GRN. However, the authors do not compare with only a learnt residual in each layer while ignoring all previous residuals ($w_1 x_t + w_2 f(x_t)$). Would this already be sufficient?
2. Table 2 presents results where improvements with DCA diminish with increasing width. Can the authors estimate the rank in each layer to verify if similar trends like Fig 5 extend to an experimental setting.
3. An analysis of the importance of layers given the learnt weights of DCA is missing in the experiments. I believe this would be crucial to highlight the differences between standard residual connections and DCA. Are there specific layers that obtain a larger weight in residual connections of DCA and if so which are they?

While the overall mechanism proposed with GRN is simple and can help achieve improved performance, it is unclear which of these layers enabled with residual connection are most important and how the optimal $k$ can be estimated for at task.

**Questions For Authors:**

See above

**Relation To Broader Scientific Literature:**

The paper discusses residual connections and attention, two crucial mechanisms in modern deep learning.

**Theoretical Claims:**

The authors show that the proposed GRN model can achieve lower risk.

---

> ### Author Rebuttal · Authors · 2025-03-31
>
> We thank the reviewer for their time and insightful feedback on our manuscript. We address the three questions raised below.
>
> > Most efficiency gains seem to occur by using the first and last-k layer outputs in the GRN, for k=2. Moreover the perplexity gains from increasing k further are limited. This implies that not all layer representations contribute to the residual connection in GRN. However, the authors do not compare with only a learnt residual in each layer while ignoring all previous residuals (w1xt+w2f(xt)). Would this already be sufficient?
>
> The method suggested by the reviewer has previously been proposed as LAuReL-LR.
> The LAuReL-PA method that we used in our experiments is a stronger generalization of LAuReL-LR, as observed in the LAuReL paper. Thus, we opted to report LAuReL-PA in our manuscript instead. Based on our results, including previous outputs in addition to the last output leads to significant perplexity improvements.
>
> We also conducted experiments without the model inputs as an explicit input to each GRN but we found that this did not perform as well as including the model inputs (num_layers=6, emb_dim=512 on lm1b):
>
> | Method   | Perplexity   |
> | -------- | ------- |
> | Transformer   | 20.878   |
> | GRN (last 4)  | 20.301 |
> | GRN (model inputs + last 3)  | 20.227 |
>
> We would like to emphasize that even with k=2 all the layer representations do contribute to the residual connections in GRN. This is because all the intermediate layers are summed, as in a vanilla residual network, and passed as an additional input to the GRN. With k=2 each GRN thus has the following 4 inputs: the input to the model, the sum of all intermediate layer outputs, the second last layer output, and the last layer output.
>
>
> > Table 2 presents results where improvements with DCA diminish with increasing width. Can the authors estimate the rank in each layer to verify if similar trends like Fig 5 extend to an experimental setting.
>
> Since the models used in Table 2 incorporate nonlinear activations, the appropriate notion of rank is the bottleneck rank as described in Section 4.4. In our case the bottleneck rank is the same as the width of the model because the model strictly improves as the width increases, which indicates that the model is not able to represent the same function with smaller width. Our empirical results thus align with our theoretical results because in both cases the benefit of our method decreases as the rank (or bottleneck rank) of the model increases.
>
> > An analysis of the importance of layers given the learnt weights of DCA is missing in the experiments. I believe this would be crucial to highlight the differences between standard residual connections and DCA. Are there specific layers that obtain a larger weight in residual connections of DCA and if so which are they?
>
> In Appendix H we provide insights into the importance of each layer by plotting the learnt weights for each GRN in Figure 10. The results show that the first and last few layers are most important as they obtain the largest weights. This insight led us to the design of the more efficient k-DCA which uses the model inputs and the last-k layer outputs together with the sum of all intermediate layer outputs as the input to the GRN.
>
> Moreover, we observed in the training dynamics that the model input is important for all the hidden layers, especially in the beginning of model training. We further verified this by removing the explicit connection to the model input which performed notably worse. We will enrich the discussion in Appendix H with these insights.
>
> We hope that these responses adequately address the reviewer's concerns. We believe these clarifications and additional results strengthen the manuscript.

---

> > ### Comment · Reviewer_zWQ6 · 2025-04-02
> >
> > I thank the authors for clarifying the use of model inputs and the rank of each layer. I will keep my score.

---

### Official Review · Reviewer_nMe5 · 2025-03-13

**Overall Recommendation:** 3

**Summary:**

The paper introduces DeepCrossAttention (DCA), a new mechanism that stores and uses intermediate features of transformers. DCA enables learnable, input-dependent weights to mix preceding intermediate features, enhancing the model's representation power. The authors also provides theoretical justifications regarding why DCA blocks have higher representation powers compared to standard ResNet architectures. Experiments on LM1B and C4 show that DCA achieves competitive performance compared to vanilla Transformer architectures.

**Claims And Evidence:**

Most of the claims are supported by clear and convincing evidence.

**Essential References Not Discussed:**

No.

**Experimental Designs Or Analyses:**

I have checked the soundness/validity of the experimental designs in the paper.

**Methods And Evaluation Criteria:**

The proposed methods and evaluation criterion make sense for the problem.

**Other Comments Or Suggestions:**

1. Could the authors provide analysis regarding the memory usage during forward and backward pass?
2. Can the authors extend their comparison to include baselines with larger-scale models to assess the method's effectiveness at higher parameter counts?

**Other Strengths And Weaknesses:**

Strength:
1. The paper is well-written and easy to read and follow.
2. The paper provides theoretical justifications alongside relatively large scale training experiments to validate the proposed method.

Weaknesses:
1. The majority of the paper is comparing DCA with vanilla transformers. And the comparison with baseline methods in Table 5 is limited to very small number of parameters, i.e., ~50 M. The authors should extend their analysis to larger-scale models.
2. There is no discussion regarding memory requirements during the forward and backward passes. It would be beneficial for the authors to report the maximum memory usage in both phases to better check the method's scalability.

**Questions For Authors:**

No other questions.

**Relation To Broader Scientific Literature:**

The main contribution of the paper is that it proposes a way to use a small number of additional parameters to improve the performance of the LLM.

**Theoretical Claims:**

I did not check the proofs of theorems in the paper.

---

> ### Author Rebuttal · Authors · 2025-03-31
>
> We thank the reviewer for taking the time to review our manuscript. We hope that our responses adequately address the reviewer's concerns. We believe these clarifications and additional results strengthen the manuscript.
>
> > Could the authors provide analysis regarding the memory usage during forward and backward pass?
>
> Since the difference in the number of model parameters is negligible, the main difference in memory usage comes from the number of activations that need to be stored.
>
> Let us first analyse the memory usage during inference. The vanilla transformer only stores one activation tensor which each transformer block adds to in order to compute the residual connection. DCA takes all the previous layer outputs as its input, thus the number of activations it needs to store scales linearly with the depth of the model, which significantly increases the memory usage for deep models. To mitigate the memory overhead, we propose k-DCA, which only uses the model’s input, last-k layer outputs, as well as the sum of the remaining intermediate layer outputs. This reduces the number of stored activations to k+2 times that of the vanilla transformer, which is independent of the model depth.
>
> During model training the memory usage of DCA is on the same order as the vanilla transformer. This is because the vanilla transformer also needs to keep all activations in memory to compute the gradients. As the DCA layer acts in a modular way on its input activations, the forward and backward passes can be performed by storing only one additional activation tensor. Based on profiling a 179M parameter model, we see that the peak memory footprint increases from 5.3GB to 6.6GB (this can likely be lowered further since our implementation is not tuned for efficiency). We also note that DCA is compatible with model sharding, which can greatly reduce the memory consumption per chip. Finally, DCA does increase the memory bandwidth because for each layer we need to read the activations of all other layers instead of just one. This additional cost is already factored into the runtime analysis presented in the manuscript which shows that DCA obtains lower perplexity for a given training time.
>
> >  Can the authors extend their comparison to include baselines with larger-scale models to assess the method's effectiveness at higher parameter counts?
>
> Based on the reviewer’s suggestion, we conducted additional experiments with larger-scale models to compare our method against related work. We have started additional runs comparing DCA against all baselines at 179M parameters (n_layers=13, emb_dim=1111, hid_dim=4444) on the C4 dataset. While our experiments are still running, we can provide the following perplexity results for the models at 400K (out of 500K) steps.
>
> | Method    | Perplexity, 400K steps |
> | --- | --- |
> | Transformer  | 21.772   |
> | 1x1-DenseFormer    |  21.313  |
> | Hyper-Connections   |  21.261  |
> | LAuReL-PA |  21.117 |
> | 8-DCA (ours)   | **20.625**   |
>
> The preliminary results show that even at larger scale our method outperforms prior work. We will add the final results to the paper once all runs are complete.

---

> > ### Comment · Reviewer_nMe5 · 2025-04-04
> >
> > Thank you for your feedback. I am maintaining my rating.

---

### Decision · Program_Chairs · 2025-05-01

**Decision:**

Accept (poster)

**Comment:**

This paper proposes an alternative to residual connections in Transformers, which dynamically reweights previous layers' outputs based on input dependent weights. On language modeling experiments, it was shown that the proposed approach improves upon model convergence compared to both standard Transformer and related baselines. All reviewers agree that this is a sound idea, and appreciate its simplicity. Questions are raised mostly around the empirical side, eg lack of larger scale evaluations, tasks other than language, analysis on memory consumption etc. The authors did a pretty good in answering these, especially in providing new experimental results. Although it is still highly desirable to further flesh out the empirical side of this work, due to the generality of the claim, this work does already demonstrate promising results and I'd lean towards accepting if there is space.

One additional point, I think the authors should cite and discuss the Highway Networks paper, which is a well established prior work and shares similar design spirit.